# U–Pb geochronology documents out-of-sequence emplacement of ultramafic layers in the Bushveld Igneous Complex of South Africa

James E. Mungall[1], Sandra L. Kamo[1] & Stewart McQuade[2]

Layered intrusions represent part of the plumbing systems that deliver vast quantities of magma through the Earth's crust during the formation of large igneous provinces, which disrupt global ecosystems and host most of the Earth's endowment of Pt, Ni and Cr deposits. The Rustenburg Layered Suite of the enormous Bushveld Igneous Complex of South Africa has been presumed to have formed by deposition of crystals at the floor of a subterranean sea of magma several km deep and hundreds of km wide called a magma chamber. Here we show, using U–Pb isotopic dating of zircon and baddeleyite, that individual chromitite layers of the Rustenburg Layered Suite formed within a stack of discrete sheet-like intrusions emplaced and solidified as separate bodies beneath older layers. Our U–Pb ages and modelling necessitate reassessment of the genesis of layered intrusions and their ore deposits, and challenge even the venerable concept of the magma chamber itself.

[1] Department of Earth Sciences, University of Toronto, 22 Russell St, Toronto, Ontario, Canada M5S 3B1. [2] Bushveld Chrome Resources, Postnet Suite 911, Private Bag x153, Bryanston, Johannesburg 2021, South Africa. Correspondence and requests for materials should be addressed to J.E.M. (email: mungall@es.utoronto.ca).

arge igneous provinces are the products of massive, short-lived magma generation that produce volumes of magma exceeding 0.1 million km[3], and are expressed by the formation of volcanic sequences up to several km thick with areal extents of $>0.1$ million km[2], giant radiating mafic dike swarms, and large layered mafic and ultramafic intrusions[1]. The largest and best studied example of a layered intrusion is the Rustenburg Layered Suite (RLS) of the Bushveld Complex, South Africa (Fig. 1). There is no consensus on the details of its mode of formation, but it is generally assumed that the layered rocks represent an upward-aggrading pile of crystals deposited on the floor of a vast, long-lived and repeatedly replenished magma chamber[2–4]. This assumption is so entrenched that it has become customary to borrow stratigraphic terminology to describe structural relations between layers as resembling bedding surfaces and unconformities. However, this interpretation has been challenged by the controversial suggestion that ore-hosting ultramafic layers formed by out-of-sequence injection of thin sills into older mafic layers[5–8].

The Rustenburg Layered Suite of the Bushveld Complex is a series of layered igneous rocks exposed in five arcuate lobes spanning 450 km from east to west and 350 km from north to south (Fig. 1). Its Cr and platinum-group element (PGE) deposits constitute a major portion of the world's economic reserves[9]. Recent U–Pb geochronology suggested that the RLS was emplaced into the Transvaal Supergroup within a period of $<1$ Myr[10], which confirmed the earlier data[11–15] and improved age precision to better than $\sim 0.1\%$ (refs 10,12). Simple thermal models indicate that the entire thickness of the complex could have been emplaced and reached its solidus temperature within 0.1 Myr[10,16]. The precision and extent of existing geochronological data have not permitted a distinction between the contradictory hypotheses of a single evolving magma chamber or a sheeted sill complex.

The layers of the RLS are subdivided vertically into several zones on the basis of regional-scale variations in lithology (Fig. 1). From bottom to top of an idealized section these are: the Marginal, Lower, Critical, Main and Upper zones[17–19]. Despite the utility at the regional scale of the broad zonal classification, the layers within them show numerous and complex variations in their spatial distribution[20,21]. The Lower Zone occurs as trough-like bodies that may be entirely separate from each other (Fig. 1). Similarly, the Critical Zone occurs in several different sectors and compartments, separate from each other along strike and characterized by unique sequences of layers (Fig. 1)[20,21]. The Critical Zone is distinguished from the Lower and Main zones by the occurrence within it of chromitite layers; in the Lower Critical Zone all of the layers are ultramafic whereas the Upper Critical Zone includes noritic and anorthositic layers[3,19–23]. The Main and Upper zones are the most laterally persistent zones of the RLS, although the latter shows a regionally discordant basal contact against the underlying components and must therefore be considered a separate intrusion, younger than the other zones[7,24,25]. Despite the pronounced regional differences in the detailed sequence of layering of the Lower Critical Zone, a widely used stratigraphic nomenclature resembling a bar-code distinguishes Lower Group (LG1 to LG7), Middle Group (MG1 to MG4) and Upper Groups (UG1 to UG3) of chromitite seams, numbered in sequence from bottom to top[22]. The base of the Upper Critical Zone is defined by the lowest appearance of mafic layers, between MG2 and MG3. It is generally assumed that the layers occur in chronological order from oldest at the bottom to youngest at the top and obey the stratigraphic principle of superposition. Correlations between chromitite layers in different compartments of the Lower Critical Zone are based on their relative positions and to a lesser extent on salient aspects of their appearance and composition but without more a rigorous basis for correlation

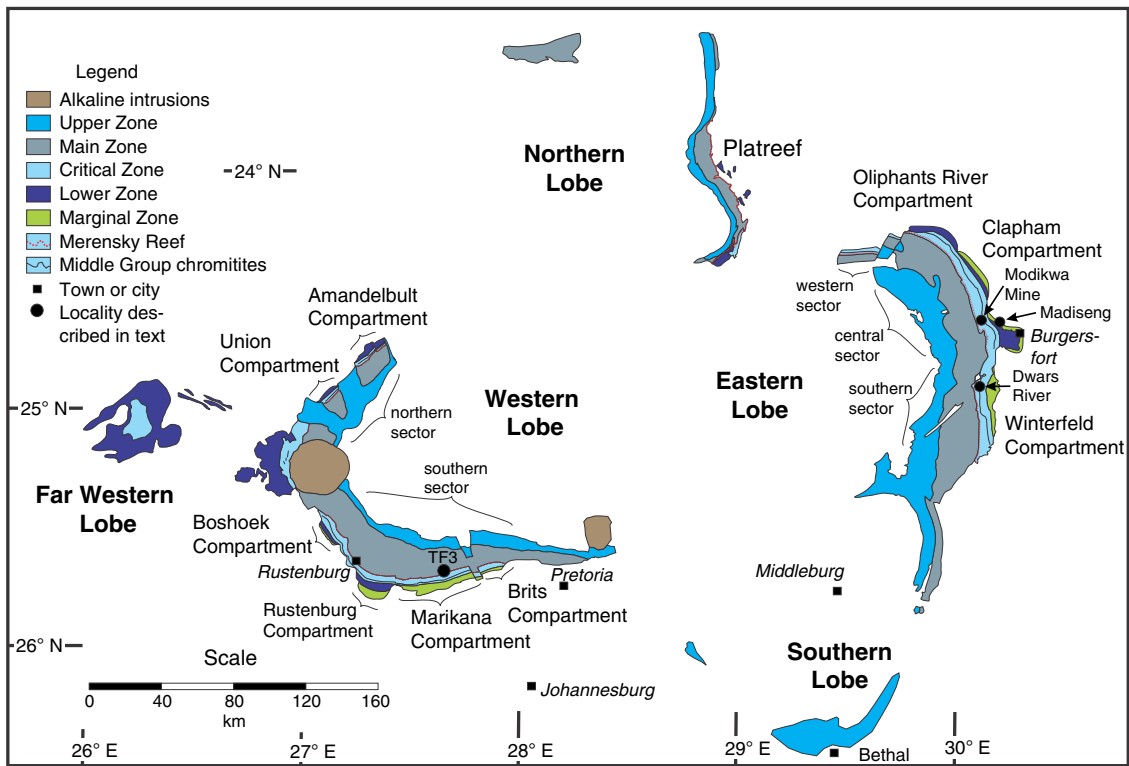

**Figure 1 | Geological map of the Rustenburg Layered Suite of the Bushveld Complex.** Compiled from multiple sources cited in the text[3,17,20–24,27]. Much of the Lower Critical Zone and all of the Lower Zone appear as discrete intrusions beneath the more continuous Upper Critical Zone and Main Zone.

there is no *a priori* assurance that a particular unit in one compartment formed from the same magma as the unit with the same name in another compartment[20,21]. The sequences of silicate macrolayers between chromitite layers differ notably between compartments[21]. In contrast, however, the uppermost units, from the LG6 through UG1, UG2, UG3 and Merensky Reef, are so uniformly exposed throughout the RLS that their respective origins from individual magmatic events are unquestioned. It is noteworthy, however, that the UG3 unit is separated from the UG2 unit by tens of metres of intervening pyroxenitic rocks everywhere north of Burgersfort, but occurs directly on top of the UG2 unit everywhere south of Burgersfort[26]. The uppermost unit in the Critical Zone includes the Merensky Reef, a major stratiform pyroxenite-hosted deposit of PGE-rich magmatic sulfide possessing only cm-scale chromitite seams[2,9,23,26]. It is generally assumed that the upper contact of the Merensky Reef is concordant, but exposures in mine workings are limited, and Mitchell and Scoon[7] noted a discordant uppermost contact of the pyroxenite portion of the Merensky Reef at Winnaarshoek. In the northern lobe of the Bushveld Complex, pyroxenite intrusions that host the PGE deposits of the Platreef, and are correlated with the Merensky Reef of the eastern and western lobes[27] are clearly seen to crosscut both the overlying Main Zone and earlier rocks of the Critical Zone[7]. The Main Zone must therefore also be considered as a separate intrusion, older than both the overlying Upper Zone and the mineralized ultramafic intrusives of the Upper Critical Zone, as has been suggested based on field relations elsewhere[28,29].

Here we present field observations coupled with high-precision U–Pb isotope dilution-thermal ionization mass spectrometry (ID-TIMS) ages of zircon and baddeleyite from the lower Main Zone and Upper Critical Zone of the RLS. Individual ultramafic macrolayers comprising chromitite and pyroxenite have sharp intrusive contacts against anorthositic rocks both above and below, and in several cases are younger than layers situated above them in the rock column. We conclude that at least some of the ultramafic macrolayers were emplaced as sills into pre-existing norites rather than having formed as deposits of crystals along the base of a large, multiply replenished magma chamber. Our data demonstrate that the RLS formed over a duration of at least 0.6 Myr between $2056.28 \pm 0.15$ and $2055.54 \pm 0.27$ Myr ago. The new data necessitate profound reassessment of commonly accepted petrogenetic models of the evolution of the contained ore deposits and of LIP magmas and layered mafic-ultramafic intrusions in general.

## Results

**Fieldwork.** The present study is based on core from the first 828 m of the vertical diamond drill hole TF3 in the Western lobe (25.692° S, 27.588° E) and on observations of the Merensky Reef and UG2 units in underground mine workings and two vertical boreholes (OV709, OV711) from the Modikwa Mine (24.64°S, 30.13°E) and the excellent exposure of the UG1 chromitite at the Dwars River UNESCO World Heritage Site (24.911° S, 30.104° E) in the eastern lobe of the RLS (Fig. 1). Figure 2 shows a schematic summary of the lithologies intersected by TF3 (summary log in Supplementary Data 1). It was collared near the base of the Main Zone and spans the entire Upper Critical Zone, terminating within the uppermost Lower Critical Zone. The hole intersected mostly norite, leuconorite and gabbronorite, totaling 730 m of these three varieties of mafic rock. From the top downwards, the Merensky Reef and UG2/3, UG1, MG4, MG3 and MG2 chromitite layers were intersected. The hole ends below the MG2 chromitite. Some of the chromitite layers appear as multiple seams (for example, MG4A and MG4B). The sequence of

chromitite layers is similar to sequences observed slightly farther east[21]. Each chromitite layer occurs within a general sequence, from bottom to top, of lower chromite-bearing anorthosite, chromitite, feldspathic pyroxenite, norite and upper mottled anorthosite (Fig. 3, Supplementary Fig. 1). In some units the norite is absent between feldspathic pyroxenite and upper mottled anorthosite; in others, instead of the lower chromite-bearing anorthosite, a mafic pegmatite is observed. In the mine exposures the mafic pegmatites are commonly observed to alternate with the chromite-bearing anorthosite along the base of the UG2 chromitite (Fig. 4). Similar sequences were observed in holes OV709 and OV711 from the Modikwa Mine (Fig. 3). For convenience herein we refer to each chromitite-pyroxenite pair as a Unit that shares the number of the associated chromitite. Another noteworthy feature of some of the chromitite layers, most notably the UG1 and MG4A, is the widespread development of chromitite and anorthosite interlayered on

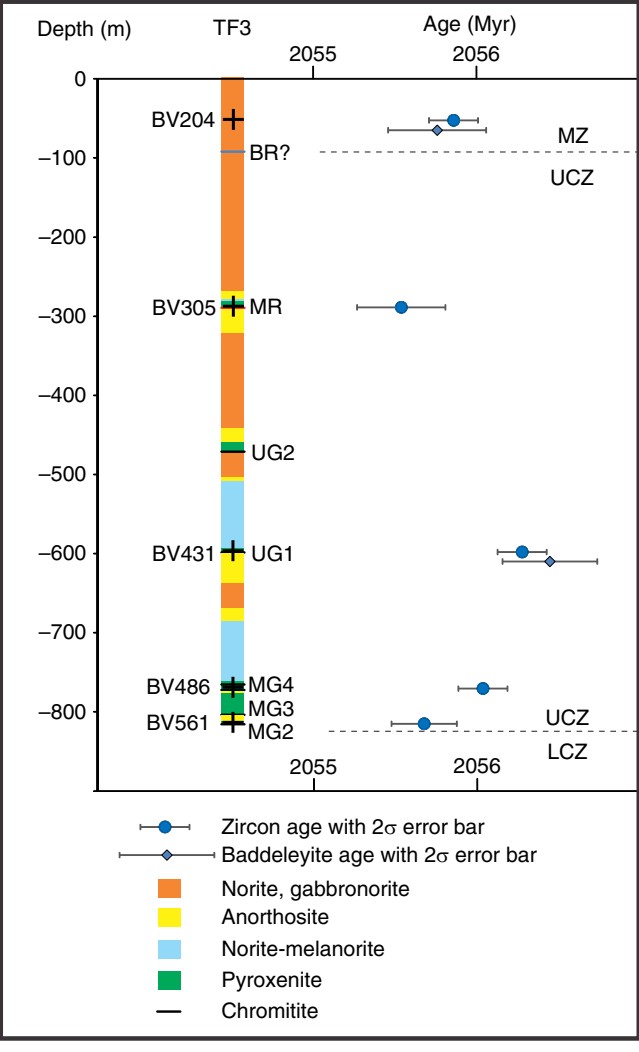

**Figure 2 | Schematic chronostratigraphy of lithologies intersected by DDH TF3 in the Western Lobe.** The colours along the borehole are a simplified representation of the more detailed lithological log in Supplementary Data 1. Weighted mean zircon $^{207}\text{Pb}/^{206}\text{Pb}$ ages determined in this study are shown (BR is Bastard Reef; MR is Merensky Reef; UG2, UG1, MG2-4 are chromitites as described in the main text; LCZ is Lower Critical Zone; UCZ is Upper Critical Zone; MZ is Main Zone). Crosses in the section refer to locations of dated samples. Error bars show 2σ confidence intervals.

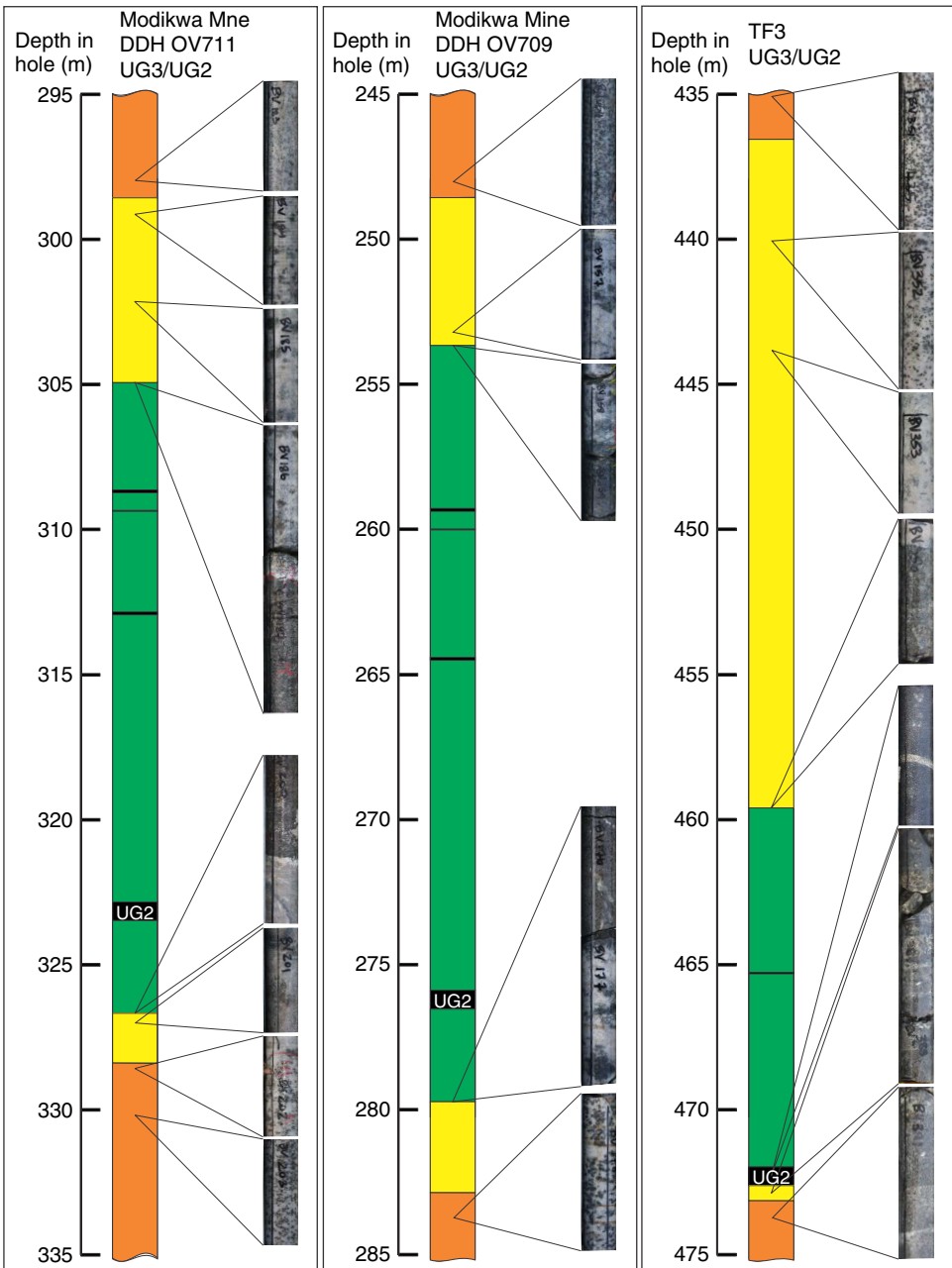

**Figure 3 | Schematic sections and detailed core photographs of key intrusive contact textures.** Contacts shown are tops and bottoms of the composite UG2/UG3 Unit, at the Modikwa Mine in the Central sector of the eastern Bushveld, and in the TF3 drill hole. At Modikwa Mine both upper and lower contacts of the UG2/UG3 Unit are sharp, separating ultramafic rock (green) from anorthosite (yellow) which grades up or down into undisturbed norite (brown); at TF3 the upper contact is sharp against anorthosite but the lower contact is occupied by mafic pegmatite.

scales from mm to dm. These layers bifurcate and locally display intense folding[23] and internal brecciation (Fig. 5a–c,e,f). Also noteworthy (Fig. 5d) is the occurrence of textures that were previously referred to as 'migmatite' and attributed to partial melting of the footwall norite in a thermal aureole below the UG1 chromitite[23]. In Fig. 5g,h, we document the occurrence of stacked sills of lithologies characteristic of the Upper Critical Zone that crop out in the Marginal Zone near Madiseng, in the Burgersfort area. This stack of sills show the general sequence (chromitite)-pyroxenite-norite-(anorthosite; Fig. 5g,h), but are situated structurally beneath the Lower Zone, well outside the conventionally defined location of the Critical Zone.

**Lithogeochemistry.** Samples of feldspathic pyroxenite, chromitite, anorthosite and norite were taken from the TF3 drill core within the Merensky Reef and the UG1, MG4, and MG2 Units and their flanking layers for lithogeochemical analysis and U–Pb geochronology. Composite samples of pyroxenite were taken from above the chromitite in each unit and a composite sample of gabbronorite of the Main Zone was also obtained. Descriptions of the rock samples dated, and detailed measurements of depth in hole of each core fragment in each composite sample, are provided in the Supplementary Note 1. Photographs of representative pieces of drill core are given in Supplementary Figure 2 and X-ray maps of polished thin sections of each dated sample are given in Supplementary Figure 3. Selected

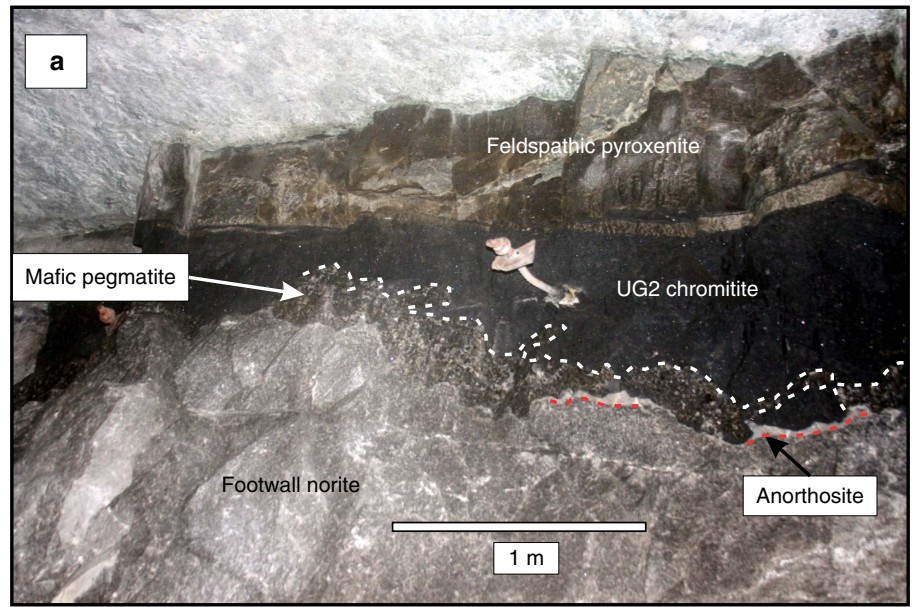

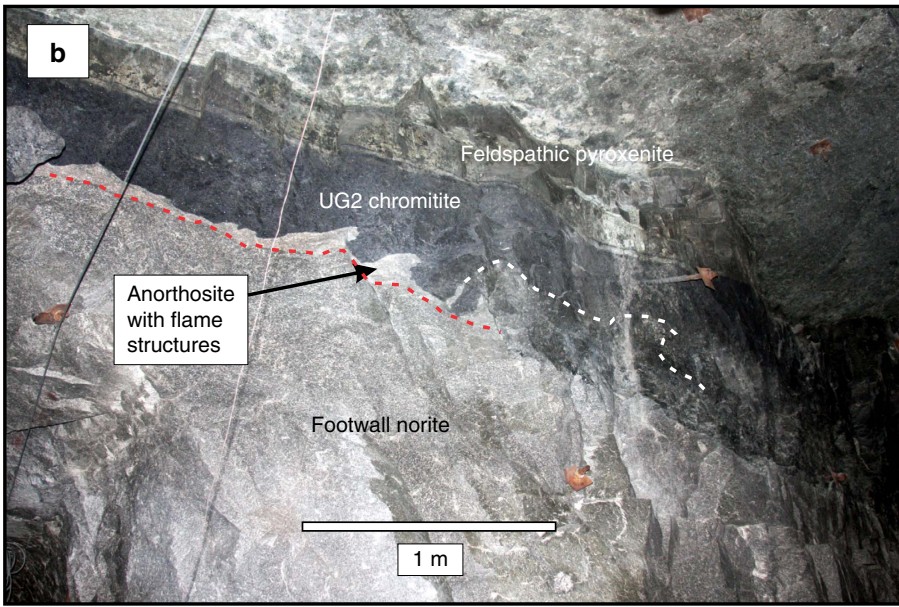

**Figure 4 | Underground exposure of the base of the UG2 chromitite (black) at Modikwa mine.** This exposure is in a pothole environment without lower pyroxenite. Contacts between anorthosite and underlying norite are emphasized with dashed red lines. Contacts between mafic pegmatite and chromitite are emphasized with dashed white lines. Scale bars are 1 m. (**a**) The contact is lined with anorthosite (white) or with mafic pegmatite. Randomly placed drill holes can be expected to encounter one or the other unpredictably. The mafic pegmatite is rooted on the footwall norite but forms irregular masses penetrating deep into the overlying chromitite. (**b**) Anorthosite shows flame structures against overlying chromite, suggesting incipient gravitational instability that could be stretched into sheath folds by layer-parallel shearing during sill emplacement.

lithogeochemical data and detection limits are reported in Supplementary Data 2.

**Geochronology**. U–Pb analyses were performed at the Jack Satterley Geochronology Laboratory in the Department of Earth Sciences, University of Toronto, by ID-TIMS methods[28–32]. Highly precise weighted mean $^{207}Pb/^{206}Pb$ ages with 2σ errors of ∼0.01% or better can be achieved with multiple, reproducible, and concordant U–Pb data that have been corrected internally for mass fractionation of Pb and U. The EARTHTIME project has made available to the U–Pb community common isotopic tracer (or spike) solutions that have been precisely calibrated[28]. The use of such tracer solutions has significant advantages. It has eliminated a major source of bias among U–Pb ID-TIMS

laboratories that use the tracers and will permit more confident age comparisons and correlations from data produced by different laboratories. A key analytical advantage of the ET2535 tracer is the direct measurement of Pb and U fractionation during analysis, such that the isotopic ratios can be corrected within each measurement cycle. This removes the largest single source of analytical uncertainty in U–Pb ID-TIMS analyses and its use has therefore led to enhanced age precision.

U–Pb isotopic data obtained on chemically abraded zircon[29] from the five samples, and baddeleyite from two of the samples, are reported in Supplementary Data 3 and the results are plotted on Concordia diagrams in Fig. 6. Single crystals or up to five fragments of zircon grains were selected for each U–Pb analysis. VG Sector software was used for data acquisition. U–Pb data

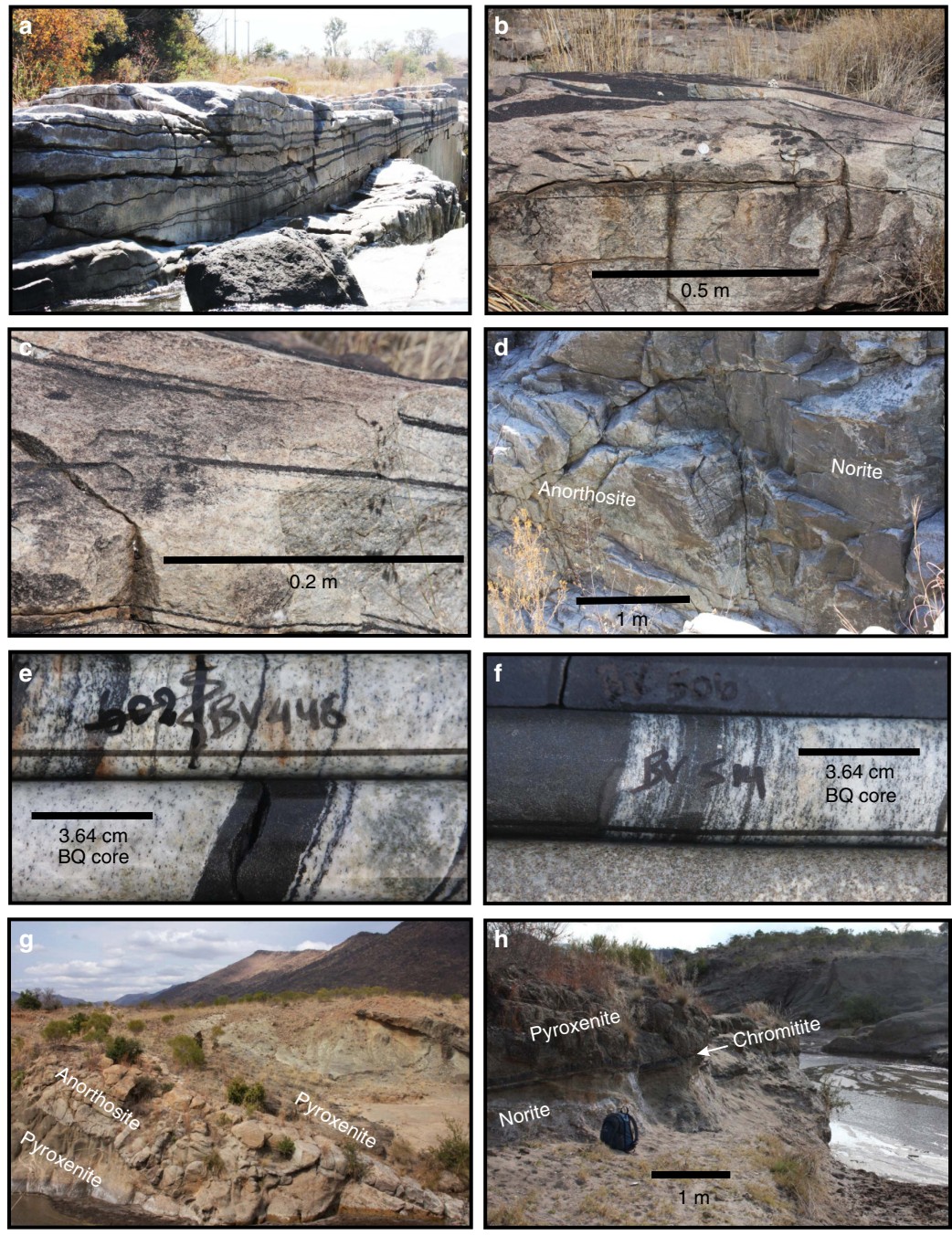

**Figure 5 | Photographs of key contact relations.** (**a–d**) show the lower part of the UG1 Unit and its footwall at the Dwars River UNESCO World Heritage Site; **a** shows bifurcating layers of chromitite (vertical face in foreground is about 1 m high); (**b,c**) show fragments and disrupted laminae of chromitite in anorthosite, however chromite itself commonly contains fragments of anorthosite; **d** shows norite with wispy irregular layering partially engulfed by mottled anorthosite in 'migmatitic' texture. **e** and **f** show similar interlayered chromitite and anorthosite in the TF3 borehole at the bases of the UG1 and MG4A Units, respectively. **g** and **h** are photographs from the Marginal Zone in the Madiseng area near Burgersfort. **g** shows stacked pyroxenite sills separated by anorthosite (24.633° S, 30.197° E), with the pyroxenites of the Lower Zone above in the distance. The rock face is about 3 m high. **h** shows a pair of thin chromitite seams forming the base of a layered pyroxenite-norite sill overlying norite (24.636° S, 30.202° E) a few tens of metres structurally beneath and 500 m east of the outcrop shown in **g**. Backpack is shown for scale.

reduction and age calculations were performed using in-house Visual Basic programs by D.W. Davis. Ages are reported as $^{207}Pb/^{206}Pb$ weighted means. All age errors and graphical plots are reported with 2σ analytical uncertainties using IsoplotEx[33]. A summary of the age results is presented in Table 1 and a plot of the ages versus depth in the hole is shown in Fig. 2. Our reported ages range from 2056.28 ± 0.15 Myr for the UG1 to 2055.54 ± 0.27 Myr for the Merensky Reef. The latter is slightly younger than the recently published age of 2054.89 ± 0.37 Myr for the footwall of the Merensky Reef[10]; overall our age range is similar to the span of ages from 2055.91 ± 0.26 Myr in the Marginal Zone to that of the footwall of the Merensky Reef reported in the same study[10]. Our most significant result is the resolvably older age of the UG1 Unit at 2056.28 ± 0.15 Myr, and the MG4A Unit at 2056.04 ± 0.15 Myr, compared with the younger underlying MG2A Unit at 2055.68 ± 0.20 Myr. Results

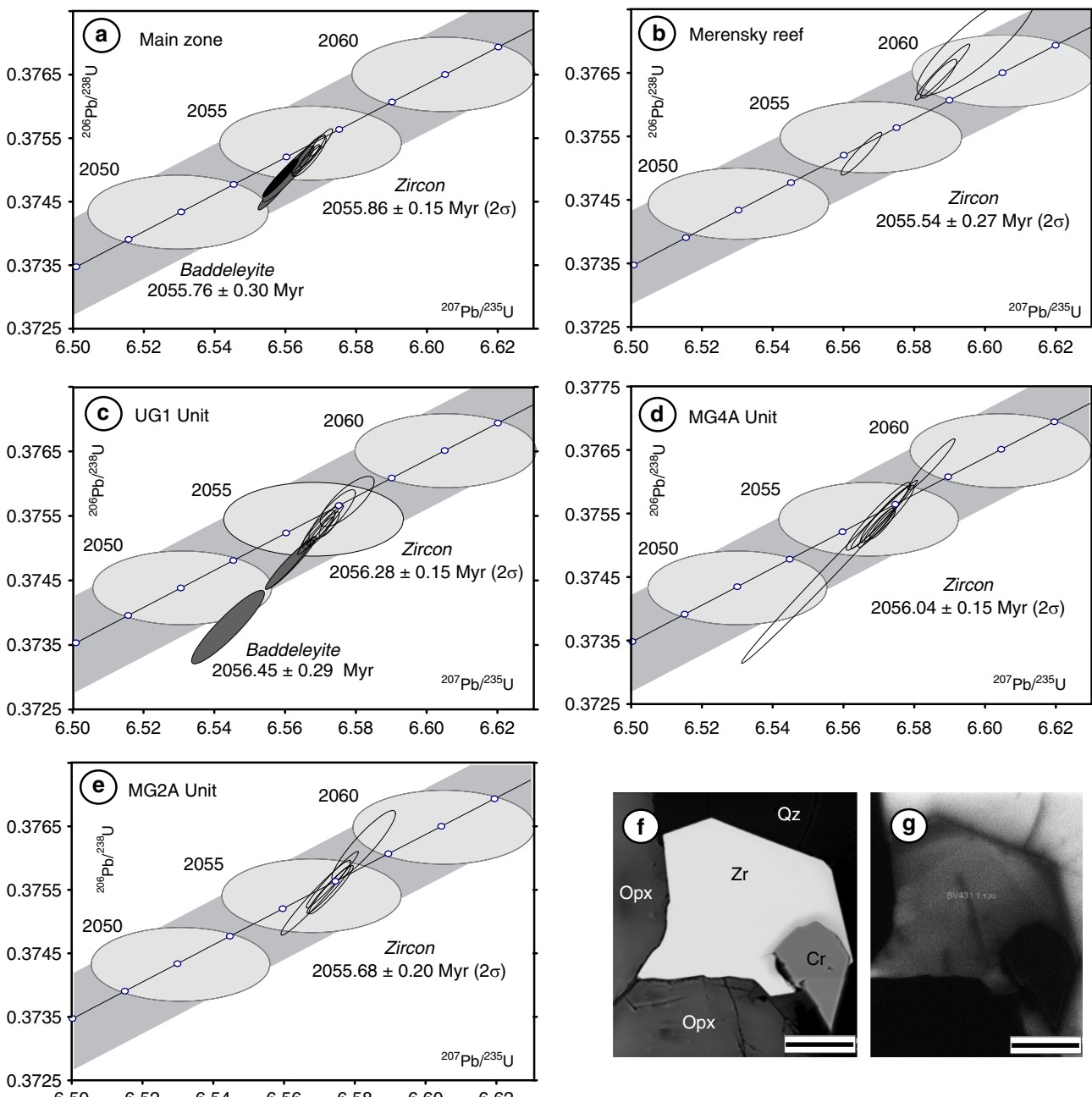

**Figure 6 | Results for five dated samples from the Main and Upper Critical zones of the Rustenburg Layered Suite. a–e** show concordia diagrams for data listed in Table 1 and described in the text. Hollow ellipses represent U-Pb results for zircon; grey ellipses represent U–Pb results for baddeleyite; black ellipse in **a** represents the one slightly younger fraction in the Main Zone sample; all ellipses represent 2σ errors. The broad grey bands indicate the uncertainty in the concordia related to uncertainty in the decay constants for U isotopes; errors on individual age points along the concordia are indicated by large grey ellipses superimposed on the concordia curve. All zircon and most baddeleyite data are concordant within uncertainties in the decay constants. Quoted ages are weighted mean $^{207}Pb/^{206}Pb$ ages. It is noteworthy that the Main Zone appears to be older than the Merensky Reef, and both the MG4A and MG2A units are younger than the UG1 unit. **f** shows a backscattered electron image of typical igneous zircon (Zr) overgrowing pyroxene (Opx) and chromite (Cr), in turn enclosed in quartz (Qz) in feldspathic pyroxenite (Sample BV431, UG1 Unit). **g** shows a cathodoluminescence image of the same field of view as **f**. Scale bar, 40 μm (**f,g**).

for the Main Zone include one zircon fraction, Z6, that has a younger $^{207}Pb/^{206}Pb$ age than the others (inclusion of this population increases the mean square weighted deviation (MSWD) from 0.78 to 3.07) but is also slightly more discordant than the others (0.16%), which may indicate a small amount of ancient Pb loss leading to an erroneously young apparent age. We have calculated and presented the ages for the Main Zone both with Z6 (2055.75 ± 0.30, MSWD = 3.07; $n = 6$) and without it

(2055.86 ± 0.15, MSWD = 0.78; $n = 5$). Baddeleyite ages from both the UG1 pyroxenite and Main Zone are identical, within analytical uncertainty, to their respective zircon ages.

## Discussion

We address the statistical significance of the U–Pb zircon age differences between our samples using the $T_u$ Student's $t$ statistic

**Table 1 | Summary of U–Pb ages for rocks from the Critical and Main zones.**

| Sample no. | Rock Unit | Weighted mean $^{207}Pb/^{206}Pb$ age* | Dated mineral | N | MSWD | Weighted mean $^{207}Pb/^{206}Pb$ age† |
|---|---|---|---|---|---|---|
| BV-204‡ | Main Zone gabbronorite | 2055.86 ± 0.15 Myr | Zircon | 5 | 0.78 | 2056.65 ± 0.15 Myr ago |
| BV-204§ | Main Zone gabbronorite | 2055.75 ± 0.30 Myr | Zircon | 6 | 3.07 | 2056.55 ± 0.30 Myr ago |
| BV-204 | Main Zone gabbronorite | 2055.76 ± 0.30 Myr | Baddeleyite | 2 | 0.68 | 2056.56 ± 0.30 Myr ago |
| BV-305 | Merensky Reef pyroxenite | 2055.54 ± 0.27 Myr | Zircon | 4 | 0.29 | 2056.33 ± 0.27 Myr ago |
| BV-431 | UG1 pyroxenite | 2056.28 ± 0.15 Myr | Zircon | 7 | 0.98 | 2057.04 ± 0.15 Myr ago |
| BV-431 | UG1 pyroxenite | 2056.45 ± 0.29 Myr | Baddeleyite | 3 | 0.05 | 2057.24 ± 0.29 Myr ago |
| BV-486 | MG4A pyroxenite | 2056.04 ± 0.15 Myr | Zircon | 8 | 1.9 | 2056.84 ± 0.15 Myr ago |
| BV-561 | MG2A pyroxenite | 2055.68 ± 0.20 Myr | Zircon | 5 | 1.7 | 2056.48 ± 0.20 Myr ago |

*Calculated with $^{238}U/^{235}U$ of 137.818 (ref. 59) for comparison with dates reported by Zeh et al.[10].
†Calculated with $^{238}U/^{235}U$ of 137.88 for comparison with all other existing data sets.
‡Calculated using the preferred 5 zircon analyses.
§Calculated using all 6 zircon analyses.

for weighted means with pooled variance[34]. For each pair of ages we consider the null hypothesis $H_0$ that the zircon populations measured all came from the same normally distributed population (that is, the two samples are the same age). If $H_0$ is rejected, then we accept $H_1$, that the two samples are distinct in age. Supplementary Data 4 shows that, for each pair of weighted ages, the value of the $T_u$ statistic, the calculated number of degrees of freedom (d.f.) and the probability of the $H_1$ hypothesis that the two samples in question are not the same age. The most robust conclusion reached is that the Middle Group 4A and 2A Units were emplaced, with 92% and 99% confidence, respectively, in reverse sequence beneath the UG1 layer. Using the age of 2055.86 ± 0.15 Myr ($n = 5$; MSWD = 0.78) for the Main Zone sample BV204, we determine with 98% confidence that the Merensky Reef is 0.32 Myr younger than the overlying Main Zone. However if we include Z6 in the estimate of the age of the Main Zone its weighted mean age decreases to 2055.75 ± 0.30 Myr, and confidence that it is older than the underlying Merensky Reef falls to 84%, which does not quite meet the conventional 90% bar for rejection of the null hypothesis but is nevertheless strongly suggestive that the Merensky Reef is younger than the Main Zone and therefore might be hosted by a sill. We note that our baddeleyite results for pyroxenite from UG1 and Main Zone samples, being analytically identical to our zircon ages, provide further support for the age comparisons we have made using only the U–Pb zircon results.

Our U–Pb results for the Upper and Middle Group units show that it is not possible to accept that the macrolayering of the Upper Critical Zone resulted from the deposition of crystals along the base of a long-lived magma chamber. We propose instead that the noritic rocks of the Upper Critical Zone formed first as a sill or sills lacking conspicuous layering. Each narrow chromitite + pyroxenite ± norite Unit that we have identified represents a separate later ultramafic intrusion, separated by sharp upper and lower contacts from anorthositic margins (Figs 3 and 4). The anorthosites flanking the Units represent restite material remaining after partial melting of the enclosing noritic rocks[35] to form mafic liquids locally preserved as the mafic pegmatites but more commonly mixed into the ultramafic magmas. The ultramafic intrusions formed from mantle-derived komatiitic parental magma, which assimilated large volumes of continental crust during ascent through the lithosphere[36–39]. The highly contaminated and partially crystallized magmas were emplaced as (olivine)-orthopyroxene-clinopyroxene-chromite crystal mushes in sills at their present level of exposure, while the remaining relatively crystal-poor magma escaped from the sills to be emplaced as the irregular intrusions of the Marginal Zone.

We present two tests of our sill emplacement hypothesis. First, we have done thermodynamic modelling of the formation of the sills and their restitic margins using alphaMELTS software[40–42]

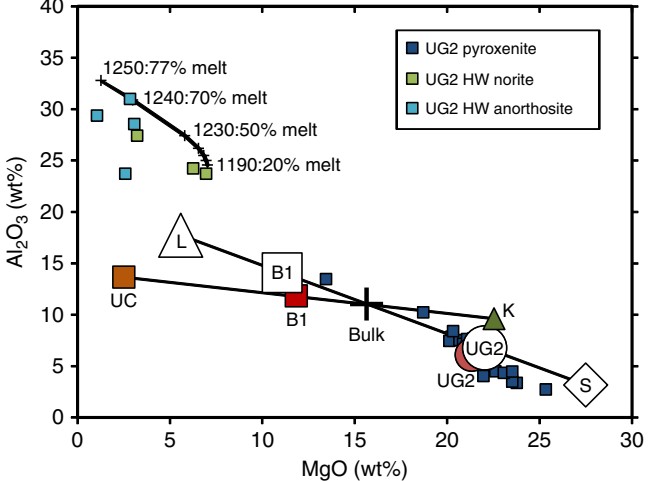

**Figure 7 | Diagram showing $Al_2O_3$ versus MgO for whole rocks and model compositions from the Bushveld Complex.** Rock compositions are shown with small coloured squares identified in the inset key (HW denotes hanging wall). Large coloured symbols: UG2 represents the average of data from the TF3 intersection through the UG2 Unit with each analysis weighted by the proportionate length of core it represents, UC is average upper crust[44], B1 is average B1 magma[37], K is komatiite[43]. The large cross is a bulk mixture of 65% komatiite with 35% upper crust, chosen as a plausible crystal-laded magma composition entering the UG2 sill. Large open symbols represent model compositions calculated using alphaMELTS[40–42] at 1,170 °C as described in the text; L is liquid, B1 is a model magma comprising 75% liquid + 25% solids, UG2 is a model cumulate composed of 25% liquid + 75% solids, S is solids. Modeled partial melting of norite in the UG2 hanging wall produced residual solid compositions shown by the curved black line at upper left. Labelled crosses on this line represent the composition of the restite at the stated temperature and degree of melting. Residues are anorthositic; mixing lines between restites and model liquid (L) would generate much of the observed range of anorthosite to leuconorite compositions in the margin of the UG2 Unit.

with bulk $H_2O$ content of 0.1 wt% and oxygen fugacity fixed at the value of the fayalite–magnetite–quartz solid oxygen buffer (that is, FMQ; Figs 7 and 8). We represent a mantle-derived ultramafic magma by selecting an uncontaminated alumina-undepleted komatiite from Thiel Well, Australia[43]. For a crustal contaminant we use the average continental crust[44]. The modeled liquidus temperature of the komatiite at 200 MPa is 1,540 °C. After assimilation such that the bulk mixture comprises 35% upper crust and 65% komatiite, the temperature of the magma has fallen to 1,235 °C. During assimilation of intermediate-felsic

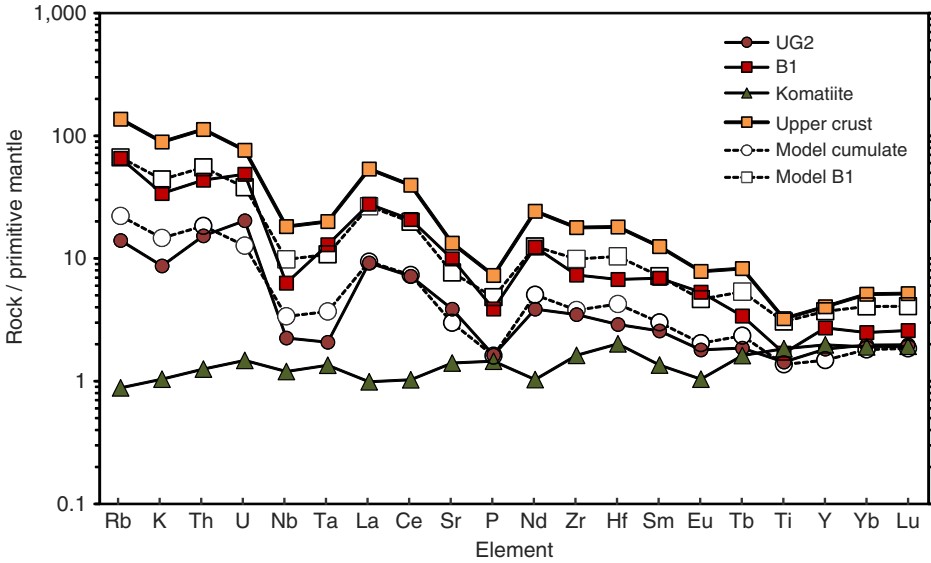

**Figure 8 | Primitive mantle-normalized trace element concentrations for natural and model rocks from the Bushveld Complex.** Solid symbols joined by solid lines represent rock analyses; upper crust[44] average B1 magma[37], weighted average of UG2 in hole TF3, komatiite sample NXMW43-3 (ref. 43). Open symbols joined by dashed lines represent alphaMELTS[40–42] models. Model cumulate is 75% solids +25% liquid; model B1 is 25% solids +75% liquid.

crustal rocks the mass of liquid remains approximately unchanged because a mass of pyroxene and spinel would form, which is approximately equal to the mass of assimilated intermediate-felsic rock. The contaminated magma comprises 63.6% liquid, 35.6% orthopyroxene and 0.7% Cr spinel. After further conductive cooling to 1,170 °C the magma comprises 53.9% liquid, 36.7% orthopyroxene, 8.0% clinopyroxene and 1.0% Cr spinel, which is in accordance with the observed primocryst assemblage in the clinopyroxene–porphyritic pyroxenites (Supplementary Figs 2 and 3). Magmas expelled into the Marginal Zone surrounding the RLS, referred to as the B1 magmas[37], resemble model mixtures of 75% liquid and 25% solids, whereas the weighted average composition of the UG2 pyroxenites from our results matches that of a model mixture of 25% liquid and 75% solids (Figs 7 and 8). Eales et al.[38] performed a modelling exercise using alphaMELTS and similar choices of magma and contaminant, envisioning formation of mushes, that resulted from crustal contamination of komatiite in the conduit, which later settled at the base of a magma chamber; analogue models have demonstrated that layering within individual units could be developed due to slurry flow along the base of a magma body[45]. We concur with some aspects of these recent proposals but propose that the mixture of melt and crystals produced by wholesale crustal assimilation in the deeper magmatic conduits was then, as suggested by Mitchell and Scoon[7,8], emplaced through narrow sills rather than at the base of a large open magma chamber and acquired its layering by slurry flow during transport. These sills extended beyond the current outlines of the RLS into the now-eroded portions of the Marginal Zone that formerly lay above what now is exposed adjacent to the Lower Zone. What is now preserved in the sills within the RLS is a basal crystal mush zone, whereas the overlying residual magma, mixed with melts derived by partial melting of the noritic wall rocks, continued to flow through the sills until it was expelled laterally as magma that eventually solidified in the Marginal Zone as pyroxenitic and noritic intrusions. It is noteworthy that the narrow chromitite seam we observed in the Marginal Zone near Madiseng (Fig. 4g,h) was deposited at the base of a sill only a few metres thick. Its genesis could not have required the operation of processes within a large magma chamber, but it might represent a

narrow distal tip of a Lower Zone chromitite-bearing unit similar to the ones we document in the Upper Critical Zone.

The amount of chromite and PGE preserved at the bases of some of the units exceeds the amount that could be derived from the observed or modeled pyroxenitic mush[46]. This implies that the chromite and magmatic sulfides were deposited from volumes of magma much greater than what remained in the narrow sills, by selective retention as heavy lag deposits, from magmas passing through the sills and exiting to the Marginal Zone. Also shown in Fig. 7 is the trend of compositions expected to result from partial melting of a norite above the UG2 pyroxenite to leave anorthositic restites, which we obtained by modelling with alphaMELTS[40–42]. At a temperature of 1,235 °C, in partially melted norite in contact with the hot contaminated komatiite flowing into the sill, the model predicts that only plagioclase should remain, generating an anorthositic restite nearest to the contact. At increasing distances and lower temperatures above the contact the degree of melting would be less, producing the series of compositions shown from anorthosite toward the original composition of the norite wall-rock. The resemblance of model restites to the observed anorthosites confirms that the anorthosite occurring both above and below each of the pyroxenite macro-layers in the Critical Zone could have been generated by partial melting of pre-existing noritic cumulate rocks. We call attention to the flame structures illustrated in Fig. 3 and note that larger-scale gravitational instability of this nature, combined with shearing along the base of a sill undergoing slurry flow[45], could generate sheath folds that would, in cross-section, resemble the complex lamination and interdigitation of chromitite and anorthosite for which the UG1 Unit is renowned (Fig. 5).

Second, we address the remarkable lateral continuity and parallelism of the ultramafic sills, features that have previously been attributed to the postulated deposition of layers of crystals on the floor of a magma chamber. Whereas it has been suggested that the magma chamber initially grew by repeated intrusion of sills before earlier sills had completely solidified[47], we suggest that the stress field beneath a sill, whether or not it has completely solidified, will guide subsequent intrusions to be emplaced below and parallel to it. We present a first-order examination of the state of stress in the RLS at some time after emplacement of a thin

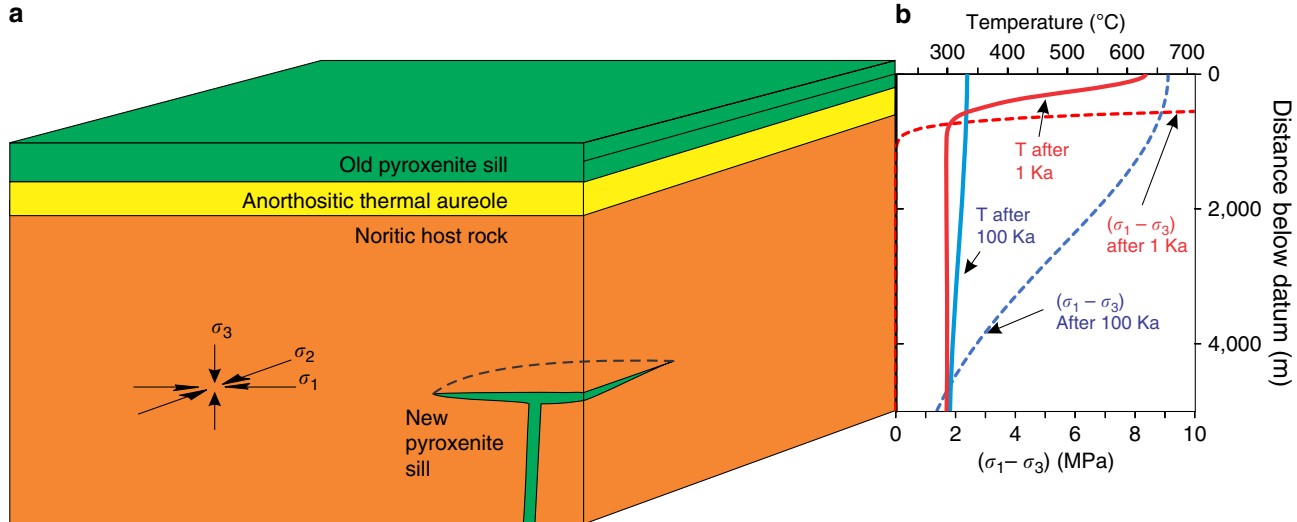

**Figure 9 | State of stress and temperature in an infinite half-space occupied by a cooling sill 200 m thick.** (**a**) shows a schematic perspective view of a cooling sill flanked by anorthosite in a norite massif, with a new dike entering from the base and being deflected into a new sill parallel to the older sill. The orientations of the three orthogonal principal stresses are shown at lower left. (**b**) shows temperature and stress deviator 1,000 years and 100,000 years following emplacement of the first sill. The first sill and its partially melted anorthosite aureole are both completely solid after <1,000 years. Even though temperature has largely relaxed to far-field values after 100,000 years, a substantial and upward-increasing stress deviator will tend to encourage subsequent intrusive dikes to deflect into sills in the $\sigma_1$–$\sigma_2$ plane (illustrated schematically in **a**). The depth of deflection will depend on the relative buoyancy of the new magma and on the magnitude of the stress deviator. Once initiated, new sills will be forced to propagate parallel to existing sills perpendicular to the minimum compressive stress direction of the deviatoric stress tensor.

sill-like body with a large lateral extent within cooler host rocks of uniform physical properties, using conventional modelling approaches to thermoelasticity[48]. To a first approximation, the Upper Critical Zone below an existing sill, before the introduction of a new ultramafic intrusion, can be modeled as a semi-infinite space at a uniform temperature whose upper surface is occupied by a horizontal hot layer with half-thickness of 100 m corresponding to the most recently emplaced body of magma 200 m thick (Fig. 9). We arbitrarily set the temperature of the host rock to 300 °C and the temperature of the first intrusion to 1,400 °C. Using higher far-field host rock temperatures would give lower far-field model stresses but has little effect on stresses close to the first sill. The datum is placed at the centre of the first sill with the vertical dimension $x$ increasing downwards. The temperature distribution $T_{x,t}$ over $x$ and time $t$, follows equation (1)[49], where $h$ is the half-thickness of the original sill and $D$ is the thermal diffusivity. For the stated geometry, with the system confined in the horizontal plane but not in the vertical dimension, the thermoelastic problem corresponds to the case of plane stress illustrated in Fig. 9. The two larger principal compressive stresses $\sigma_1$ and $\sigma_2$ are equal, horizontal, and equal to the sum of the lithostatic pressure and the calculated thermal stress (equation (2))[48], whereas the minimum principal stress $\sigma_3$ is unaffected by the thermal state and is equal solely to lithostatic pressure $P_{\text{lith}}$ (equation (3))[48].

$$T_{x,t} = T_{x=-\infty} + \frac{T_{x=0} - T_{x=-\infty}}{2}\left[erf\left\{\frac{h-x}{2\sqrt{Dt}}\right\} + erf\left\{\frac{h+x}{2\sqrt{Dt}}\right\}\right]$$
(1)

$$\sigma_1 = \sigma_2 = P_{\text{lith}} + \frac{6K\mu\alpha}{\lambda + 2\mu}T_{x,t}$$
(2)

$$\sigma_3 = P_{\text{lith}}$$
(3)

Using a Young's modulus of 20 GPa and Poisson's ratio $\mu$ of 0.25 (ref. 50), the bulk modulus $K$ is $1.33 \times 10^{10}$ Pa. With thermal expansivity of $1 \times 10^{-5}$ K$^{-1}$ (ref. 51), we find that after 1 Ka the

stress deviator (that is, $\sigma_1 - \sigma_3$) increases sharply about 300 m below the base of the cooling older sill to values greater than 10 MPa (Fig. 9). Even after 100 Ka the thermal stress is an increasing function with structural height, rising to values near 10 MPa at the base of the original sill. The model temperatures are far below those at which viscoelastic relaxation of the thermal stresses would be anticipated[52]; even if viscous responses occur in some hotter parts of the system, the simple functional dependence of the plane stress on temperature in equation (2) shows that such behaviour in hotter domains will not affect the thermal stresses in the cooler portion of the system that behaves in a purely elastic manner. Similar magnitudes of far-field stress deviators have been shown to control the direction of propagation of dikes[53,54]. The existence of a vertical minimum compressive stress is one of the key criteria that can cause a propagating dike to be deflected into a horizontal sill[50]; once the sill begins to propagate it will continue to expand laterally along the $\sigma_1$–$\sigma_2$ plane (Fig. 9), which is exactly parallel to the sill previously emplaced. Provided that a sufficient supply of magma exists, this second sill can propagate to follow the entire extent of the previous sill, at a constant spatial separation below it, guided by the stress field imposed by the existence of the earlier sill. The prior existence of a sheet of cooling magma with dimensions on the order of several hundred kilometres wide by several hundred metres thick will therefore encourage the subsequent emplacement of sills underneath it and exactly parallel to the existing layering as a simple consequence of the presence of a stressed thermal aureole underlying each successively emplaced sill. Formation of new sills beneath existing sills will be strongly favored over penetration of a propagating dike through the stressed thermal aureole and the previously emplaced sill itself, contrary to previous treatments that neglected the state of stress[47]. Although the thermoelastic model presented here oversimplifies a much more complex geological reality, it serves to show that the observed parallel disposition of ultramafic layers in the RLS is fully consistent with the physics of sill emplacement and need not be regarded as an extraordinary coincidence.

From our simple model results we anticipate a tendency for the formation of stacked sequences of sills that are emplaced at successively lower levels, younging down-structure, as is observed in the UG1-MG4A-MG2A sequence. The age differences between Units exceed their expected solidification times by orders of magnitude, precluding inflation of existing sills[47] to form a magma chamber. Published ages for the Lower Critical Zone[10] are younger than our age for the UG1 pyroxenite. The Lower Critical Zone may therefore represent a stack of sills similar to those we have dated, emplaced entirely beneath the older mafic intrusions, with ages of emplacement younging downwards due to constraints on sill emplacement caused by thermal stresses imposed by earlier sills. These sills occur in at least nine separate compartments disposed radially around the circumference of the RLS (Fig. 1), and may therefore be regarded as the type example of a giant radiating sill swarm at least 450 km in diameter, perhaps analogous to the giant radiating dike swarms commonly associated with large igneous provinces[1].

We echo the recent statement[15] that 'geochronological tools are now available to address whether major layered intrusions are assembled in discrete magmatic episodes, (and) whether all layered intrusions are simple stratigraphic sequences of cumulates with the oldest rocks at the base and the youngest at the top'. A complex history emerges for the development of the Bushveld Complex, in which noritic rocks of the Upper Critical Zone formed early, followed by the mafic Main and Upper zones, a process that was punctuated episodically by the intrusion of ultramafic sills into the slightly older but completely solidified mafic intrusions over a period of at least 0.6 Myr. Our observations and models call into question the concept of a single giant 'advancing, periodically replenished, periodically tapped, continuously fractionated' reservoir[55] (that is, a magma chamber) that remained open and active throughout the main pulse of magmatic activity[4,9,56]. Instead, the U–Pb dating results and field observations show that some ultramafic units of the Upper Critical Zone were emplaced as sills into pre-existing mafic rocks and tend to support previous suggestions that the Merensky Reef is younger than the Main Zone above it[6–8]. Petrogenetic models for chromitite and PGE deposition that require settling of crystals or sulfide melt from deep columns of magma[9,56] or by upward percolation of fluids through underlying partially molten cumulates[57] in the RLS depend on assumptions that are contradicted by our new geochronological results and must be revised to take account of the radically different intrusion geometry, or may even need to be discarded entirely. The idea that large homogeneous volumes of highly contaminated magma can be generated through monotonously repeated processes of assimilation along the walls of magma conduits instead of requiring the existence of a well-mixed magma chamber[58] also is supported by our present findings.

Our proposal that macrolayering in a layered intrusion might result from the emplacement of a series of individual sills at successively lower levels need not be restricted to the Bushveld Complex but could also be applied other well-known layered intrusions. The existence of intrusive upper contacts on the sills is very easy to overlook due to the absence of recognizable chilled margins on these hot erosive boundaries as illustrated in Supplementary Fig. 1.

The notion of the 'magma chamber' is so deeply ingrained in current thinking about igneous petrology that it is invoked in countless papers without any attempt at justification. Here we have shown that much of the layering in one of the world's truly iconic layered intrusions need not be explained in terms of processes occurring in an open melt-filled chamber. Invisible and undetected 'deep-seated magma chambers' and 'staging chambers' appear throughout the current literature, but the notion of a staging chamber beneath the Bushveld Complex or any other complex magmatic system may also be unnecessary. The composition of many batches of incoming crystal–laden magma can be modeled simply by taking a common Proterozoic magma type (komatiite) and adding some melted wall-rock to it as it rises through the continental crust along a flow-through conduit[58]. We therefore urge some cautious reflection before appeal is made to the prevalent concept of vast open chambers filled with essentially crystal-free melt except in those cases where direct evidence can be seen for their existence. Although small magma chambers like the famous Skaergaard Intrusion of east Greenland[18] have doubtless existed at some times and places, their primacy as the model for formation of large layered intrusions must now be critically re-examined in each case.

## Methods

**Lithogeochemistry.** Major element oxide and trace element concentrations of the core samples were determined at ActLabs of Ancaster, Ontario. Samples were crushed in steel jaw-crusher and pulverized in an agate puck mill before fusion in lithium metaborate/tetraborate flux using an induction furnace. The molten solutions were poured into a 5% nitric acid solution containing an internal standard and completely dissolved. The resulting aqueous solutions were analysed using Thermo Jarrell-Ash Enviro II inductively coupled plasma mass spectrometer (ICP-MS), which was calibrated using multiple USGS and CANMET certified reference materials (WMG-1, NIST 694, DNC-1, GBW 07113, MICA-FE, CHR-BKG, CHR-PT, AJ-G, BE-N, UB-N, NIST 1633b, LKSD-3W-2a). Loss on ignition (LOI) was determined gravimetrically before and after heating to 1,000 °C in air.

**Geochronology.** U–Pb geochronology was done at the Jack Satterley lab at the University of Toronto. Rocks were crushed and pulverized using standard methods with a jaw crusher and disk mill. A Wilfley table was used to produce a concentrate of heavy minerals. Zircon and baddeleyite were isolated using methylene iodide and a Frantz magnetic separator. Before dissolution, zircon grains were thermally annealed and chemically etched to penetratively remove alteration zones where Pb loss has occurred[29]. Grain weights were estimated from photomicrographs. These zones correlate with areas of high U that have suffered radiation damage before alteration. To thermally anneal damaged lattice sites, grains were placed in a muffle furnace at ∼1,000 °C for up to 60 h. This was followed by partial dissolution in 50% HF and ∼10 μl 8 N $HNO_3$ in Teflon dissolution vessels at 195 °C for 12–14 h. The grains were washed in 8 N $HNO_3$ before dissolution. A mixed $^{202-205}$Pb–$^{233-235}$U spike (ET2535 tracer from the EARTHTIME Project[28]) was added to the Teflon dissolution capsules during sample loading. The zircon and baddeleyite were dissolved using ∼0.10 ml of concentrated hydrofluoric acid (HF) and ∼0.02 ml of 8 N nitric acid ($HNO_3$) at 195 °C (ref. 30) for 5 days, then dried to a precipitate, and re-dissolved in ∼0.15 ml of 3 N hydrochloric acid (HCl). U and Pb were isolated from the zircon solutions using anion exchange chromatography, then dried in dilute phosphoric acid ($H_3PO_4$), and deposited onto outgassed rhenium filaments with silica gel[31]. U and Pb were analysed with a VG M354 mass spectrometer either in static mode using multiple Faraday cups or dynamic mode with a Daly pulse-counting system. During static measurements (with $^{207}$Pb, $^{206}$Pb, $^{205}$Pb, $^{202}$Pb in Faraday cups), the $^{204}$Pb beam was measured in parallel in the axial Daly detector. Daly gains were calibrated and updated before each analysis. Faraday cup amplifier gains were measured and updated daily by the use of constant current sources to a precision of about 10 p.p.m. Corrections for isobaric interferences from $BaPO_{16}O_{17+}$ on $^{202}$Pb were made by measuring ratios of $^{202}$Pb to $^{201}$Pb before and after each static Pb measurement. Corrections for isobaric interferences from $^{233}UO_{16}O_{18}$ on $^{235}UO_2$ at mass 267 have been made. The dead time of the Daly measuring system for Pb and U was 16.5 and 14.5 ns, respectively. The mass discrimination correction for the Daly detector is constant at 0.05%/atomic mass unit. Daly characteristics were monitored using the SRM 982 Pb standard. Thermal mass fractionation was measured and corrected within each cycle for both Pb and U. The total common Pb in each zircon analysis was attributed to laboratory Pb (corrected using an isotopic composition of $^{206}$Pb/$^{204}$Pb of 18.49 ± 0.4%, $^{207}$Pb/$^{204}$Pb of 15.59 ± 0.4%, $^{208}$Pb/$^{204}$Pb of 39.36 ± 0.4%; 2σ uncertainties), thus no correction for initial common Pb from geological sources was made. Routine testing indicates that laboratory blanks for Pb and U are usually less than 0.5 and 0.01 pg, respectively, but common Pb can be introduced during analysis. Zircon and baddeleyite U–Pb measurements were made over the course of 2 years during which time aliquots of the EARTHTIME 2 Ga synthetic solution[28] were analysed periodically. Collectively, these data have a weighted mean $^{207}$Pb/$^{206}$Pb age of 2000.78 ± 0.23 Myr (MSWD = 1.13; N = 6). Corrections to the $^{206}$Pb/$^{238}$U and $^{207}$Pb/$^{206}$Pb ages for initial $^{230}$Th disequilibrium have been made (excluding data from synthetic solution analyses) assuming a Th/U ratio in the magma of 4.2, based on assumed crustal average and on the model liquid values in Supplementary Data 2. Decay constants are those of Jaffey et al.[32] ($^{238}$U and $^{235}$U are 1.55125 × 10$^{-10}$ and 9.8485 × 10$^{-10}$ per year, respectively). All age errors quoted in the text and tables, and error ellipses

in the Concordia diagrams are given at 2σ. The error associated with spike calibration on the U/Pb ratio for ET2535 is reported at <0.05% (95% confidence[28]).

**Data availability.** The authors declare that all relevant data are available within the article and its Supplementary Information files.

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

## Acknowledgements

J.E.M. was supported by a Discovery Grant from the Natural Sciences and Engineering Research Council of Canada and by endowed funds associated with the Norman Keevil Chair in Ore Genesis Geology. Access to drill core and mine workings was provided by Samancor Chrome and Anglo American Platinum.

## Author contributions

J.E.M. and S.M. performed field work and logging of drill hole TF3. S.L.K. acquired U–Pb geochronological data and wrote related text. J.E.M performed modelling and wrote the manuscript. All authors discussed the results and edited the manuscript.

## Additional information

**Competing financial interests:** The authors declare no competing financial interests.

