## [Peer Review File · Nature Communications]

Reviewers' comments:

Reviewer #1 (Remarks to the Author):

General Comments

This is a very interesting contribution on one of the landmark intrusions of petrology, the Bushveld Complex, a mafic-ultramafic layered intrusion that has influenced many of our basic concepts of how igneous bodies crystallize and how magmatic ore deposits of chromium, platinum group elements, and vanadium form. The authors report new U-Pb zircon dates for rocks from the Critical Zone and Main Zone of this intrusion, the largest mafic-ultramafic layered body on Earth. Based on a statistical treatment, they use relatively small differences in age between the samples to propose that the constituent layers do not form in a regular fashion from bottom-to-top, but are in fact out-of-sequence, which implies that some/many layers were emplaced as sills into pre-existing cumulates. Evidence of emplacement as sills is based on inferred field relationships (provided as photographic evidence) and the textural-chemical fingerprint of partial melting of host gabbro-norite that produced very thin residues of contact anorthosite. From this, the authors speculate on the physical emplacement mechanisms of these sills, which may extend laterally for 100s of km, and provide a simple thermomechanical model to explain how new sills should follow pre-existing sills and how each successive sill should be emplaced beneath consolidated sills. The end result is a view of the emplacement and crystallization of layered intrusions that is markedly different from the conventional view of progressively crystallizing magma chambers, a view that is slowly emerging from the study of the Bushveld Complex and other large mafic-ultramafic layered intrusions throughout the geological record.

The strength of the study lies in the deft combination of high-precision U-Pb dating, field relationships, and simple modeling results. The U-Pb geochronological results are excellent and will be extremely useful to future workers as they continue to unravel the complexities presented in dating of the Bushveld Complex. There is a great deal more dating that will need to be done on rocks of the Bushveld Complex to fully appreciate the timescales of emplacement, crystallization, and cooling - with only the results for 5 samples presented in this study, focused almost entirely on chromitite-bearing horizons, there are significant gaps in sampling from an intrusion that locally exceeds 8000 metres in thickness. The alphaMELTS modeling of partial melting of host gabbro-norite to produce an anorthosite residue nicely ties observations to a predictable model that can be tested with more detailed studies in the future. There is an unresolved issue with their suggestion that the partial melt mobilized from along sill margins somehow accumulates >1000 metres below crystalline rock to form the Marginal Zone of the Bushveld Complex. The thermomechanical modeling of sill emplacement provides a simple, first-order resolution to the apparent observations of decreasing age with increasing depth in the Upper Critical Zone (although this is really only relevant to the lowermost 200 metres of the UCZ, a subtlety that is not explicitly addressed). There are issues here with the actual geometry of layers in the Bushveld Complex compared to the simplified planar geometry required by the model, but these will most certainly be addressed by future workers.

The significant weaknesses of the manuscript are many and include aspects of the proposals, the writing and use of grammar, the quality and utility of the figures, and the information in the tables and supplementary files. This manuscript unfortunately needs a great deal of help to ensure that the message and the approach are clear to the reader, specialists and non-specialists alike.

Specific Comments

Page 1

- line 2: need to indicate "Bushveld Complex" in the title as this is what the community knows, not "Rustenburg Layered Suite".
- line 11: the part of the sentence beginning with "which" is incorrect as it suggests that LIPs host these major deposits when you mean that layered intrusions do.
- line 12: layered intrusions host Cr, PGE, and V deposits, but rare Ni deposits (unless we are going back to the decades old definition of Sudbury as a layered intrusion?).
- line 13: avoid using acronyms in the abstract that is written for the non-specialist - delete RLS; also state "Bushveld Complex".
- line 14: I think "ocean" is the wrong analogy - that's a small ocean, maybe "sea" is better.
- line 15: how were the chromitite layers dated? Add "uranium-lead geochronology" here.
- lines 17-19: nice statement (I totally agree!).
- line 21: use of "lavas" is incorrect, should be "volcanic deposits/flows/rocks"
- lines 24-27: nicely stated.

Page 2

- line 32: incorrect use of hyphens - should be "isotope dilution-thermal ionization mass spectrometry" to match the ID-TIMS acronym.
- lines 33-34: need a "that" after "sills", but that means 3 x "that" in one sentence - rewrite.
- line 34: use "Ma" rather than "Myr" for millions of years or millions of years ago - see GSA Today overview by Paul Renne from a few years back.
- lines 40-41: incorrect grammar - chromitite is not an element as the sentence implies.
- lines 38-41: this is not a paragraph, just two related sentences. Join with short paragraph below for a real paragraph.
- line 42: does recent geochronology (Zeh et al., 2015 EPSL) really show that the entire RLS was emplaced within 1 Ma or less? There are no samples from the Main Zone and significant gaps (1000s of metres) throughout the sequence.
- line 43: "...confirming and refining earlier lower-resolution data [references 10-15]" - what exactly does "lower-resolution data" mean? Did the previous studies not have enough samples, the wrong samples, the wrong techniques? What constitutes "high-precision U-Pb geochronology"? The authors are effectively arguing that only their results and those from Zeh et al. (2015) are "high-precision". The current definition of "high-precision geochronology" from Schmitz & Kuiper (2013 Elements, p. 26) is as follows:

"High-precision geochronology relies on isotope dilution thermal ionization mass spectrometry (ID-TIMS), a technique that can provide individual crystal-fragment and weighted mean isotope ratios (and interpreted ages) of {plus minus}0.1% and {plus minus}0.03% relative precision, respectively."

Some clarification on what the authors mean by their "high-precision" dating and everyone else's "lower-resolution data" is in order.

- line 45: "datasets" not "data"

- line 47: "single evolving magma chamber" - who said this? "radiating sheeted sill complex" - where did this come from?

- line 48: should be "zone" when on its own, not capitalized - make this correction throughout the manuscript.

- line 49: should be "bottom-to-top" with hyphens

- line 50: zone is not capitalized when in the plural form (same as formations, units, etc.); correct this throughout the manuscript.

- line 50: delete "however", not needed

- line 51: "defy simple categorization" - I don't know what this means and checked the references to try to find out too.

- line 56: add "...that contain abundant plagioclase" to help the non-expert.

- lines 57 and 58: use of the word "component" - I don't know what this means? Has it been defined? There must be clearer terms to use.

- paragraph lines 48-60: most "Zones" should be "zones".

- line 59: as argued here, the Upper Zone is the youngest part of the RLS, but that is contradicted later in the manuscript? Need some clarity here.

Page 3

- line 61: give abbreviations for the different chromitites as soon as you use them - Lower Group (LG), etc.

- line 63: the UCZ did not fall (or walk, run, jog) - use "is defined by"

- lines 67-69: I wrote "can be tested with geochemistry" on my marked up copy here - does this statement invalidate all existing geochemical studies of the chromitites?

- line 72: this is the first time I have seen the use of "interburden" - I know what it means now because I looked it up and found that it is a mining term, but I suggest replacing it with a word that most readers will understand.

- lines 73-74: multiple use of "everywhere" - avoid repetition and rephrase one part of the sentence.

- line 77: yes, Bushveld Complex, but everywhere else it is written as RLS?
- lines 77-79: awkward sentence - try rephrasing. The subject gets lost. Avoid the word "clearly" as it assumes the reader intuitively knows what the authors have observed.
- lines 77-86: this is all one paragraph - right now, these are two small incomplete paragraphs.
- line 82: there is a problem here. The authors indicate that the Platreef and the Merensky Reef are one and the same based on apparent correlations across the Bushveld Complex. This is, however, far from accepted. The packages contain distinctively different rock types. I am unaware that clear continuity has been determined between the Northern Limb and the other lobes, both from geological and geophysical evidence. Field relationships defined for the Platreef are not relevant to the Merensky Reef.
- line 89: need to provide location information for these two boreholes (lat/long, depth drilled, location on map, etc.)
- line 91: friends are "encountered" in a bar, lithologies are not "encountered" by a drill hole.

Page 4

- line 92: replace "bottom" with "base"
- line 93: replace "entirety of" with "entire"
- line 94: replace "encounter" - see line 91 above.
- line 94: as written, the sentence indicates that the gabbronorite is 730 m in thickness - a small rewrite with clear this up.
- line 95: where are the other chromitites? A small comment here might be useful in the context of these being sills.
- line 97: "... is comparable with similar profiles from slightly farther east" - I'm not sure what this means, what profiles? Just a re-wording needed.
- line 106: replace "interbedded" with "interlayered"
- line 107: replace "millimetric laminations to decametric beds" with "that range from mm to dm".
- lines 108-109: I cannot tell which photo this is? The reader needs some help here (even the specialist does, too). The notion of partial melting has not yet been brought up, so this catches the reader by surprise.
- lines 110-113: this is not a paragraph, just a single sentence that shows up out of the blue.
- line 112: delete "clearly"
- line 114: start with "Whole rock"
- line 118: replace "lithochemical data" with "geochemical results"

- line 121: hyphen needed after "dilution"

- line 122: add "weighted mean" before "207Pb/206Pb ages"

- lines 114-119: there is something critical missing here for the reader and for future workers - we need a clear indication of what was sampled and analyzed for U-Pb geochronology? Where exactly was each sample taken from with respect to each chromitite? In the middle, above, below? Are the dating samples actually chromitites or the rocks in the package? This needs to be spelled out in the supplementary information with short statement here. I suggest indicating how far away individual samples from upper and lower contacts of chromitites were taken in centimeters. Even better - a photograph of the 5 samples (drill core) with a box indicating what was sampled.

Page 5

- line 123: "multiple, reproducible, concordant U-P data" is awkward - try a rewrite.

- line 134: add "... and the results are plotted on..." at the end of the sentence.

- line 135: "single crystals or up to 5 fragments of zircon grains" - need to be very clear about this in the data table (add a column), which specific analyses are single grains and which are multi-grain fractions? Are all baddeleyite analyses done on single grains as well?

- line 136: "207Pb/206Pb weighted means" is incorrect - should be "weighted mean 207Pb/206Pb dates"

- line 137: is there any specific data collection or data reduction software used in addition to Isoplot?

- line 137: should be "age results", not "data" - the data are in the analytical table.

- line 139: ages don't "fall", people do (or branches from trees).

- line 139: need to add "Ma" after each date.

- line 139-140: "in good agreement with the recent published age range" - I don't know what this means? What constitutes "good agreement"? It looks like they may be a little younger - certainly the Merensky Reef ages do not overlap.

- line 143: replace "data" with "results"; need "sample" after Main Zone

- lines 143-145: awkward phrasing - try a rewrite. The sentence reads that the age is discordant, but that is incorrect - the U-Pb results from this one fraction are discordant. I also can't tell which fraction this is on Fig. 5A? Try a different shading.

- line 147: "to ensure rigorous evaluation of the data" - I think this is overstated, you are just reporting two different age calculations.

- lines 133-148: there is no mention of the baddeleyite dates - are these not useful or important? And I guess there are some rutile dates somewhere too based on the title of the data table?

- lines 133-148: the U-Pb results are excellent - very fine work that will be very useful for future

workers. What about the reversely discordant results for the zircon (single grains or multi-grain fractions?) from the Merensky Reef sample? Everything else behaves so well. These are different - why?

- line 149: should be "U-Pb zircon age differences"; also, it's not just between the units, it also includes the Main Zone?

Page 6

- line 155: I don't understand the definition of the "the most robust conclusion" here - the statistical analysis does not provide conclusions, just what is permissible.

- line 157: replace "our age" with "the age"

- line 159: replace "Myr" with "Ma"

- lines 156-161: no spaces before %, should be "90%"

- line 162: "but is nevertheless strongly suggestive" - suggestive of what? This is an unfinished statement.

- line 163: replace "in agreement" by "identical within uncertainty"

- line 164: replace "the zircon data" with "the U-Pb zircon results"

- line 165: replace "Our data" with "Our U-Pb results"; replace "impossible" with "not possible"; add "that" after "accept"

- line 166: delete "as being the" and replace with "resulted from the"

- line 167: replace "suggest" with "propose"

- lines 170-172: well-phrased and clear statement.

- lines 172-173: "the ultramafic intrusions formed due to assimilation, within deep conduits, of continental crust by komatiite magma" - as written, this is incorrect and misleading. The ultramafic intrusions formed from the crystallization of komatiitic magma, derived by high extents of partial melting of the mantle, which was contaminated by continental crust during transport from the mantle to the uppermost part of the crust.

- line 177: delete "types of first order" - unnecessary words.

- line 177: "our hypothesis" - which specific hypothesis? There are a number of proposals in the preceding paragraph, however, it is unclear which one is being referred to?

- line 178: what were the water contents and redox states of crystallization and partial melting? These need to be stated explicitly either here or in the supplementary table to allow researchers to reproduce the results.

- line 179: delete "generic" - not necessary

- line 180: add "upper continental crust" - I got confused here as continental crust is intermediate in composition, not felsic
- line 182 and 185: repetition of "comprises" - find another word for one of them or re-arrange a sentence
- line 183: magmas don't fall, ski, hop, or run - the temperature decreases
- line 183: "felsic crustal rocks" - the average silica content of upper continental crust is ~66 wt%, so it's better to refer to "intermediate-felsic" here; note that this composition needs to be included somewhere in Supp Table 2 for reference.
- line 184: replace "which" with "that" as the word modifying "mass" and not "forms".

Page 7

- line 185: not sure "felsic" is appropriate here for reasons stated above.
- lines 185-187: the decimal points being quoted can't be significant - just round off for the larger numbers.
- line 187: replace "matching" with "..., which matches".
- line 188: avoid using "clinopyroxene-porphyrific pyroxenite" - not a good usage for an ultramafic cumulate; delete "well" - not necessary.
- line 189: add B1 magma composition to Table 2 for reference and comparison.
- line 190: replace "data" with "results"
- lines 191-192: "performed a similar exercise" is awkward - rephrase.
- line 193: incorrect usage of "which" - here it modified "conduit", but I don't think the conduit settled anywhere!
- lines 195-198: well-stated and clear.
- lines 198-201: expelling of melt to form Marginal Zone - interesting proposal, but the implications are that there are being expelled over 100s of km strike length, unless you develop a method for forcing them downwards through crystalline rock. You have introduced a mechanically challenging process in trying to solve another. Why not just mix the melt back into the slurry during sill emplacement?
- line 202: start new sentence with "This means that the chromite..." to avoid a very long and difficult to follow sentence.
- line 203: where did the number "25 times greater" come from? It shows up without any context or support.
- lines 205-210: this is very complex and convoluted sentence that needs to be broken into pieces and reassembled as two separate sentences.

- line 211: "flame structures" - why do they have to have a sedimentary origin? Why could they not result from the proposed partial melting process?

- line 215: delete "must"

- line 216: replace "which" with "that"

Page 8

- line 217: "repeated intrusion of sills before earlier sills" - what sills are we talking about here? This is in direct reference to one paper, but many other authors have proposed non-sill origins for the Bushveld Complex.

- line 220: replace "using" with "and use"

- line 221: "thermoelasticity" of what?

- line 221: "Upper Critical Zone" - this is modeled as cold rock (573 K) - what happens when it is warmer?

- line 222: what is a "magmatic pulse" - humans have a pulse, but do magmas?

- line 224: delete hyphen between "recently emplaced" - no hyphen when the word ends in "y".

- lines 228-233: too long and complex - difficult for the reader to follow. Break into smaller bites.

- lines 221-240: this entire section where the details of the modeling are explaining could be moved to a supplementary file, which would allow the authors to expand on the geological significance and issues of emplacing these sills (or to produce a shorter article). More space is used on the background to the modeling here than on the U-Pb geochronology, which is the foundation of the study.

- line 238: replace "Kyr" with "ka"

- line 239: sentence is convoluted here with no punctuation - try putting brackets around "(200 m thick)" or re-arrange sentence.

Page 9

- lines 248-252: clear statement - critical for the reader!

- lines 253 and 256: repetition of "strongly" - find another word for one of them.

- lines 262-264: reference to younger ages reported in Zeh et al. (2015 EPSL) - yes, but their ET 2Ga synthetic solution result is 1 Ma younger than what your results give?

- line 262: there are no published ages for samples from the Lower Zone in Zeh et al.

- line 262-263: "numerous isolated compartments" - no, all the Zeh et al. samples from the CZ appear to be from one compartment in the northeast Eastern Limb with the exception of one UG2 sample from the Western Limb.

- lines 267-271: good statement, however, I think the modeling results need to address some of the realities of the layered sequences in the Bushveld Complex. The stress model involves lateral expansion parallel to the σ_1 - σ_2 plane. The layered rocks are the Bushveld Complex (RLS) are not necessarily flat, but curved with layered packages pitching out around dome structures that formed during emplacement and cooling of the hot cumulates (see excellent papers on structural geology of emplacement of the Bushveld Complex by Clarke et al 2005 SAJG, 2009 Lithos). The proposed mechanism is "simple", which is always appealing, but needs some modification in light of the geology.

- line 272: add "Bushveld Complex" here

Page 10

- line 278: replace "data" with "U-Pb dating results"

- line 279: delete "tend to" - not necessary

- lines 280-281: "PGE deposition" - the PGE (as elements) are actually deposited?

- lines 284-287: when I read this statement, I expected such a wide-reaching statement to be supported by many studies - consider other supporting references.

- lines 280-289: good statements and strong finish. I couldn't help but notice that a similar passage was written at the end of a recent review paper on the geochronology of layered intrusions in the "Layered Intrusions" book published by Springer (see Chapter 1, p. 59-60):

"By combining these advances in technologies and applications with our current capabilities of determining high-precision crystallization and cooling ages from a range of rock types in layered intrusions, Archean to Cenozoic, the next 10-15 years of research will undoubtedly provide many important, and unexpected, discoveries based on the geochronology of layered intrusions with implications for the evolution of mafic magmatism in the Earth's crust. The geochronological tools are now available to address whether major layered intrusions are assembled in discrete magmatic episodes, whether all layered intrusions are simple stratigraphic sequences of cumulates with the oldest rocks at the base and the youngest at the top near the roof, whether discontinuities in cumulate sequences can be identified and time gaps measured, and whether their associated mineral deposits are directly related to the crystallization of their immediate host rocks or formed or were modified at some temporally resolvable time after crystallization. Answers to these questions will directly impact our understanding of the emplacement, crystallization, and cooling of these exceptional bodies of igneous rocks."

An acknowledgement that others are working on this fascinating and fruitful avenue of science appears to be warranted.

References

Good use of references in general.

- is reference #8 to Scoon and Mitchell at the LASI Conference an abstract? If so, it cannot be used.

- starting on page 13, lots of journal names are not italicized - double-check these.

- need superscripts in Jaffey reference (line 403)
- need total pages for book references in #49 and #50
- the Muller & Pollard reference is cited twice (#53 and #54)

Methods

- line 497: should be "major element oxide"
- line 502: need to state what these reference materials are - either here or in the table. Hopefully some of them are mafic rocks?
- I assume that LOI was not determined by ICP-MS?
- line 504 needs to begin with "For U-Pb geochronology, ..." as this is new analytical section
- line 535: great to see a reporting of the EarthTime 2 Ga synthetic solution. I'm surprised that you make no comment about the difference between your results and those from Zeh et al. (2015) - there is a 1 Ma difference between them, which is significant given the age differences you are working with. In the text, you indicate that the two studies give similar results, but you don't address this bias.
- line 537: why a specific value of Th/U magma of 4.2? I think I know, but the non-specialist (anyone who does not know the details of Bushveld geochemistry) will not.
- line 538: give the numerical values for the decay constants here in addition to the reference. This makes it very clear what was used - just add them in brackets.

Tables

- there is no explanation in the text why the two different values of $^{238}\text{U}/^{235}\text{U}$ are used and reported in this table. If no mention is made, then why report them? To my knowledge, the international geochronology community has not yet agreed to make the shift to the Hiess et al. (2012) value of 137.818 - even these authors do not recommend accepting it without further study. Here's a quote from Blair Schoene's excellent overview on "U-Th-Pb Geochronology" in the Treatise on Geochemistry (second edition) that outlines the current thinking:

Schoene (2014 Treatise on Geochemistry), p. 359:

"Using the value of 137.818 {plus minus} 0.045 suggested by Hiess et al. (2012) would change the calculated λ_{235} by $\sim 0.03\%$, though, as discussed in that paper (Figure 13), a single study with full traceability to SI units needs to be carried out before new values of λ_{235} are adopted for use in geochronology. Furthermore, increasing the absolute resolution of U-Pb geochronology beyond the 0.1% level and/or verification of the accuracy of the λ_{238} value from Jaffey et al. (1971) requires further counting experiments."

The authors have presumably used the lower value of $^{238}\text{U}/^{235}\text{U}$ as this is what Zeh et al. used (2015 EPSL). For comparison with all existing U-Pb datasets, the 137.88 value is still the reference value. Some clarification of the table and the protocol used are needed to avoid confusion. I recommend reporting dates in the manuscript with the 137.88 value and report the dates with 138.818 values for reference in the table if you wish (maybe in supplementary material?), but make

this clear to the reader.

- under Rock Unit, flip the text around so that the unit is first, then the rock type (makes more sense).
- the title of the table is incorrect - it is not "for rocks from the Critical Zone", but for rocks from both the Critical Zone and the overlying Main Zone.
- the use of the word "population" in the footnote is incorrect. As used, the authors are indicating that there are multiple populations of zircon in their samples that they analyzed - this is not the case (I hope!). They are "analyses" or "results".

Figures

Figure 1

- need to add "Bushveld Complex" in the caption title.
- need to indicate sources for the figure - the current statement is insufficient as it assumes that the reader can figure out which references are appropriate on their own.
- the caption needs to indicate what the ages are - weighted mean $^{207}\text{Pb}/^{206}\text{Pb}$ dates.
- the drill holes for this study need to be clearly indicated on the map - I see TF3, but not the other ones.
- the various boxes (around legend, around scale, around lat/long, around middle of nowhere in lower-centre) are distracting and none of them need to be there.
- in the Eastern Lobe, the font size for the different sectors is not the same and should be.
- in the legend, I cannot tell the difference between the "Merensky Reef" and the "Middle Group" - same colour and patterns basically. And there is no indication what "Middle Group" refers to in the caption.
- the graph to the right should have tickmarks along the bottom axis to allow the reader to easily read off the ages of the lowermost samples.
- the caption needs to define the abbreviations used in the graph (BR, MZ, UG, MG) - only the specialist knows these.

Figure 2

- I like the idea of this figure, but I must admit that it is hard to see much in the scanned cores (these are great, but at this scale, I only see some colour variations and not the "key textures" as indicated). I'm not sure this will survive any further reduction on the printed page. Perhaps focus on one part and expand the scale or simplify the figure, minus the scans, and include the scans in the supplementary material?
- the decimal zeroes can all be deleted (e.g., 295 rather than 295.00) as they are not significant at this scale and take up space.

- in the caption, delete "detailed" and "key textures" as these are not viewable by the reader.
- commas are missing throughout the caption

Figure 3

- great photos, but there are some issues. First, add need a thick black line around each photograph to make them pop out of the page. Second, the photos look very washed out - the colour settings and contrast needs to be reset (this can be clearly stated in the caption). Third, I can follow the different contacts (just!), but I don't think a non-specialist can, so I recommend trying a version where the major contacts are carefully traced with thin white lines.
- the white boxes around the text in both panels are distracting - try without the boxes and use white font all around (increase size a little to make it contrast enough).
- the figure caption is poor. The caption needs a correct title as in "Photographs of ...". The sentence beginning is "Randomly placed drill holes..." is very awkward as written and hard to decipher - it needs a complete rewrite. The text for (B) does not provide the clear description that the reader needs to evaluate the photo.
- the location (level) of the mine in which these photos were taken needs to be stated.
- there is no scale given for either of these photos - a scale must be given for all photographs otherwise they are of little use to the reader.

Figure 4

- I suggest adding boxes around them to make them pop - this is especially important when the colour in the photo is nearly the same as the page (see panel A).
- the labels (A, B, C, etc.) are too large and the white boxes too distracting. Try using small white font.
- the caption needs to start with "Photographs of ...".
- Dwars River is not spelled correctly. In South Africa, it is written as "Dwarsrivier", but the usage should be in the English form here.
- the caption for A is a fragment of a sentence that does not make sense.
- line 592: delete "also"
- line 593: "interlayered", not "interbedded"
- line 598: spell out "metres" and finish the sentence (beneath G?)
- after looking at the photos and reading the caption, I realized that it will be very difficult for the non-specialist to actually see what the authors are referring to in the photo. I suggest labelling the photos (could use rock name abbreviations?) to make this clear, otherwise they are just grey rocks with some

bands and swirls.

Figure 5

- I recommend copying and pasting the tickmarks to the opposite sides of the graphs - this really helps the reader determine the values for points/symbols that are near all sides of the graphs.
- capitalize "Reef" in panel B.
- I suggest labeling the CL image as panel G (it looks me a few re-reads to get this).
- the title of the figure is incomplete: "U-Pb concordia diagrams showing...".
- the use of the word "data" is incorrect in this caption - the data are in the table, these are the results plotted on the diagrams.

Figure 6

- copy the tickmarks to the opposite sides of the graph for reference.
- the title is incorrect and should state "Diagram of Al₂O₃ vs. MgO showing ...".
- "data" are not shown - they are in the table. It should read "Whole rock compositions are shown..."
- the box in the upper left is too large and distracting - reduce the space between the three labels and it will look better.
- I have a lot of trouble reading this caption and understanding how the diagram works - I should be able to do this with just the figure and the caption.
- what does "UG2 is weighted average of TF3 intersection" mean? I think I know, but it's not clear.
- it's not a "black curved trend", it's a "curved black line".
- the last statement does not make sense to me. Two of the whole rock compositions plot on the restite trend, so they cannot be derived by mixing. Two of the whole rock compositions plot outside the possible mixing vectors and so are inconsistent with the proposed model?

Figure 7

- copy the tickmarks to the opposite sides of the graph for reference - need tickmarks for the elements along the x-axis.
- tighten up the spacing between the labels in the legend - it will look better and less crowded in the upper right.
- title is incorrect, should be "Primitive mantle-normalized trace element concentrations for ...".
- line 618: delete "actual", substitute with "rock analyses"

Figure 8

- the "old sill" text covers a line - find a way to avoid this.
- the first sentence in the caption is awkward, especially "...sill 200 m thick 1,000 years and 100,000 years..." - too many numbers. Try a rephrasing to help the reader.
- consider changing the temperature from K to {degree sign}C - intuitively simpler.

Supplementary Data Figure 1

- in the caption, depths in the drill hole need to be given for each core in the photo - don't make the reader try to find this information in the associated supplementary material
- there are no scales again - absolutely need scales.

Supplementary Data Figure 2

- need depths indicated in the caption for each sample shown

Supplementary Material

Supplementary Information - sample descriptions:

- as a stand-alone document, this needs to convey some additional information to the reader (some of which is in the main text), including location of the drill hole, total depth, angle, and width of core.
- need a hyphen in "a medium-grained..."
- change "faint igneous lamination" to "weak lamination" - presumably defined by the orientation of plagioclase crystals? Or not?
- "orthocumulate texture"
- replace all usages of "intercumulus" to "interstitial" to avoid any future issues of interpreting terminology.
- when using "the rock is fine grained ...", there is no hyphen - see other usages on page

Supplementary Table 1:

- needs a careful edit and some rewriting. As presently written, the Descriptions look like what was written on first observing the core - some tightening up of language would be useful as this table will be archived and available to future workers
- watch out for capitalization issues in headings
- Lower CTC column heading is not defined (what does this mean for the reader?)
- Unit Name abbreviations are not defined.

- some measurements in metres have two decimal places, others have none - need to be consistent.
- please avoid the \$ sign for sulphides - use "S"
- under Colour Codes, what does "mafic interburden" mean?

Supplementary Table 2:

- title: what kind of "compositions"?
- need to indicate what reference materials were used?
- lines 62-67: where do these values come from? Need to indicate how they were calculated?
- decimals issue in "Depth in hole" column
- need to indicate in footnote what methods were used to determine the major element oxide and trace element concentrations?
- decimal usage issue throughout table - what is shown is what the authors are indicating is significant, but it is variable?
- column Q, Cr₂O₃ in wt%? There are values for leuconorites to anorthosites that are 81%, 405%, 96% - these are not possible? 405%? Move the Cr₂O₃ column to the left to just after Al₂O₃ where it belongs.
- column R, Cr (ppm): need to remove decimal places in lines 22-24 and 40-46 (not significant).
- in the REE, watch out for significant decimal places (some apparently have none).
- decimal places in Yb-U under model compositions are all wrong - need to correct.
- line 59, column 2: I don't understand the label (Thiel Well; Ta, Sr, U interpolated) -make a clearer statement in a footnote

Supplementary Table 3:

- the title says that there are rutile analyses, but none are reported? I assume these will be part of a follow-up paper?
- there are no U and Pb concentrations reported? These need to be included.
- there is no mass of the fraction reported?
- footnotes should clearly indicate the decay constants used (even if they are in the main text), as well as data reduction programs, and a reference to Mattinson (2005) for the chemical abrasion pretreatment protocol. The supplementary tables need to work as stand-alone items for future reference.
- why is the assumed Th/U in the magma a very specific value of 4.2? I think I know, but it would be useful for the reader to know.

- the footnotes need some kind of symbology to reference specific parts of the table - very useful for the non-specialist.

- a separate column needs to be added to indicate how many grains were used for each fraction.

Supplementary Table 4:

- title should include "Bushveld Complex", so that future users will know what this table refers to.

- there are column headings missing (columns B and C)

- footnotes need symbols to guide the reader.

- it's not clear what the different sub-sections of this table are? For the reader to navigate the calculations, each needs a clear heading

- what does "using Z1-Z5..." mean? I know, but definitely not the non-specialist who is trying to figure out how these important calculations were made.

Reviewer #2 (Remarks to the Author):

This is, by and large, a well-written paper describing new geochronologic data, aimed at establishing age relations between different layers of the Bushveld Intrusion (or intrusions). The goal is to test a hypothesis that these different layers might be formed by a succession of discrete intrusions rather than sequential formation from a large chemically evolving magma body. Generating age data of sufficient precision to be able to distinguish the crystallisation ages of different layers in such old rocks is presumably quite a big technical challenge. It certainly looks (figure 1) as though some "age discrepancies" are present (figure 1) although perhaps one could debate whether this is enough evidence alone to support the hypothesis. However, these data are also supported by a number of field / textural observations which plausibly mostly support the interpretation being proposed. I have not seen the rocks myself in the field, although I've read about them. It is obviously a difficult job to summarize in a short paper the particular observations in an overall context of a huge amount of published work on the Bushveld. I did not always find it easy to keep track of the various descriptions of the rock units which are not immediately familiar to me (lines 48-86, for example). Finally, the paper develops the hypothesis in two quite worthwhile ways 1) modelling the melting and mixing relations from the point of view of chemical composition and mineralogy, and 2) some thermomechanical modelling to underpin the observational work with some physical ideas.

To summarize the paper is therefore quite ambitious. By and large it is very dense - there is quite a lot to assimilate given the short length. The thermomechanical modelling alone, for example, one could easily imagine developed as a paper in its own right, rather than as here, where it goes by a little fast, to be frank. These ideas seem generally plausible to me, but to be really convinced, I's probably rather see something more detailed. Overall I found the paper rather dense, but nevertheless, the different kinds of reasoning and measurements also give a quite a nice balance to the approach.

I am basically supportive of publication with no significant changes. I don't know whether it will be found convincing by all readers who know a lot more about the Bushveld than I do, and who may be

inclined to evoke counter arguments / observations, but I found it sufficiently thought-provoking and well argued enough to be a valuable contribution.

Reviewers' comments:

Reviewer #1 (Remarks to the Author):

General Comments

This is a very interesting contribution on one of the landmark intrusions of petrology, the Bushveld Complex, a mafic-ultramafic layered intrusion that has influenced many of our basic concepts of how igneous bodies crystallize and how magmatic ore deposits of chromium, platinum group elements, and vanadium form. The authors report new U-Pb zircon dates for rocks from the Critical Zone and Main Zone of this intrusion, the largest mafic-ultramafic layered body on Earth. Based on a statistical treatment, they use relatively small differences in age between the samples to propose that the constituent layers do not form in a regular fashion from bottom-to-top, but are in fact out-of-sequence, which implies that some/many layers were emplaced as sills into pre-existing cumulates. Evidence of emplacement as sills is based on inferred field relationships (provided as photographic evidence) and the textural-chemical fingerprint of partial melting of host gabbro-norite that produced very thin residues of contact anorthosite. From this, the authors speculate on the physical emplacement mechanisms of these sills, which may extend laterally for 100s of km, and provide a simple thermomechanical model to explain how new sills should follow pre-existing sills and how each successive sill should be emplaced beneath consolidated sills. The end result is a view of the emplacement and crystallization of layered intrusions that is markedly different from the conventional view of progressively crystallizing magma chambers, a view that is slowly emerging from the study of the Bushveld Complex and other large mafic-ultramafic layered intrusions throughout the geological record.

The strength of the study lies in the deft combination of high-precision U-Pb dating, field relationships, and simple modeling results. The U-Pb geochronological results are excellent and will be extremely useful to future workers as they continue to unravel the complexities presented in dating of the Bushveld Complex. There is a great deal more dating that will need to be done on rocks of the Bushveld Complex to fully appreciate the timescales of emplacement, crystallization, and cooling - with only the results for 5 samples presented in this study, focused almost entirely on chromitite-bearing horizons, there are significant gaps in sampling from an intrusion that locally exceeds 8000 metres in thickness. The alphaMELTS modeling of partial melting of host gabbro-norite to produce an anorthosite residue nicely ties observations to a predictable model that can be tested with more detailed studies in the future. There is an unresolved issue with their suggestion that the partial melt mobilized from along sill margins somehow accumulates >1000 metres below crystalline rock to form the Marginal Zone of the Bushveld Complex.

This comment stems from a misunderstanding of our proposed sill emplacement mechanism, which we have endeavored to clarify in the revised manuscript. What we suggest is that the sills were open at their ends, where they reached the limits of the Bushveld Complex, and thereby were able to intrude the metasedimentary host rocks of the Complex to form the multiple bodies of the Marginal Zone. Partial melt derived from thermal effects on the floors and roofs of the sills within the Bushveld Complex was mixed into the sill magma except in minor examples where it was preserved as small transgressive bodies of mafic pegmatite.

The thermomechanical modeling of sill emplacement provides a simple, first-order resolution to the apparent observations of decreasing age with increasing depth in the Upper Critical Zone (although this is really only relevant to the lowermost 200 metres of the UCZ, a subtlety that is not explicitly addressed). There are issues here with the actual geometry of layers in the Bushveld Complex compared to the simplified planar geometry required by the model, **but these will most certainly be addressed by future workers.**

The significant weaknesses of the manuscript are many and include aspects of the proposals, the writing and use of grammar, the quality and utility of the figures, and the information in the tables and supplementary files. This manuscript unfortunately needs a great deal of help to ensure that the message and the approach are clear to the reader, specialists and non-specialists alike. **We have followed almost all of the reviewer's suggestions (except for some trivial disagreements over wording) and we trust that the revised manuscript will be found to have been greatly improved.**

Specific Comments

Page 1

- line 2: need to indicate "Bushveld Complex" in the title as this is what the community knows, not "Rustenburg Layered Suite". **added Bushveld Complex to title.**

- line 11: the part of the sentence beginning with "which" is incorrect as it suggests that LIPs host these major deposits when you mean that layered intrusions do.

actually we do mean to say that LIPs host these deposit types - a LIP encompasses the lavas, dikes and layered intrusions and hence the location of the ore deposits in layered intrusions does indeed place them within the LIPS. The Bushveld Complex is classified by many as a LIP and it undeniably contains major ore deposits.

- line 12: layered intrusions host Cr, PGE, and V deposits, but rare Ni deposits (unless we are going back to the decades old definition of Sudbury as a layered intrusion?).

which is why it is important to recognize that we were indeed referring to all of the parts of LIPs, not only the layered intrusions, in which case this objection should vanish. So the original wording should stand - for example the Norilsk Ni deposits occur in the feeder intrusions within the Siberian Traps LIP.

- line 13: avoid using acronyms in the abstract that is written for the non-specialist - delete RLS; also state "Bushveld Complex". **done**

- line 14: I think "ocean" is the wrong analogy - that's a small ocean, maybe "sea" is better. **done**

- line 15: how were the chromitite layers dated? Add "uranium-lead geochronology" here. **added "using U-Pb isotopic dating of zircon"**

- **lines 17-19: nice statement (I totally agree!). thank you**

- line 21: use of "lavas" is incorrect, should be "volcanic deposits/flows/rocks" replaced with "volcanic sequences"

- lines 24-27: nicely stated.

Page 2

- line 32: incorrect use of hyphens - should be "isotope dilution-thermal ionization mass spectrometry" to match the ID-TIMS acronym. reworded

- lines 33-34: need a "that" after "sills", but that means 3 x "that" in one sentence - rewrite.

- line 34: use "Ma" rather than "Myr" for millions of years or millions of years ago - see GSA Today overview by Paul Renne from a few years back. oh ok.; fixed throughout manuscript (although I think this is a backward step that only increases ambiguity in writing about ages and time-spans - editor? should we use Ma for both time periods and ages or should we continue to denote periods of time as Myr and absolute ages as Ma?),

- lines 40-41: incorrect grammar - chromitite is not an element as the sentence implies.

reworded

- lines 38-41: this is not a paragraph, just two related sentences. Join with short paragraph below for a real paragraph.

- line 42: does recent geochronology (Zeh et al., 2015 EPSL) really show that the entire RLS was emplaced within 1 Ma or less? There are no samples from the Main Zone and significant gaps (1000s of metres) throughout the sequence. We consider that since all reported ages are within the stated 1 Ma time span, our statement is reasonable. Any dated sequence of rocks could, in principle, contain another layer that has not yet been dated and which would have an age outside the presently recognized span of ages; by the reviewer's reasoning it would never be permissible to state a range of ages for any group of rocks since it is always possible that another undetected age exists outside the stated range. We have tempered our statement by replacing "shown that" with "suggested that".

- line 43: "...confirming and refining earlier lower-resolution data [references 10-15]" - what exactly does "lower-resolution data" mean? Did the previous studies not have enough samples, the wrong samples, the wrong techniques? What constitutes "high-precision U-Pb geochronology"? The authors are effectively arguing that only their results and those from Zeh et al. (2015) are "high-precision". The current definition of "high-precision geochronology" from Schmitz & Kuiper (2013 Elements, p. 26) is as follows:

"High-precision geochronology relies on isotope dilution thermal ionization mass spectrometry (ID-TIMS), a technique that can provide individual crystal-fragment and weighted mean isotope ratios (and interpreted ages) of {plus minus}0.1% and {plus minus}0.03% relative precision, respectively."

Some clarification on what the authors mean by their "high-precision" dating and everyone else's "lower-resolution data" is in order.

We have re-written the sentence by removing "lower resolution data" and describing high precision ages as those with precisions of <0.1% .

- line 45: "datasets" not "data"

This seems like a trivial objection, and it proposes that we replace "data", a word with unequivocal meaning, with the recently coined jargon word "dataset" which does not appear in any dictionary and appears to have a meaning coincident with "data" (i.e., the plural of datum, implying the existence of a set of data).

- line 47: "single evolving magma chamber" - who said this? "radiating sheeted sill complex" - where did this come from?

The concept of the single evolving magma chamber appears in every paper ever written about the RLS with the possible exception of the 4 articles we cite as references 5-8. We have removed the word "radiating" but retain "sheeted sill complex" since it is implied by our leading statement that the RLS "contains a series of individual sills that do not follow a regularly upward-younging sequence" in the previous paragraph.

- line 48: should be "zone" when on its own, not capitalized - make this correction throughout the manuscript. Two such instances found and corrected.

- line 49: should be "bottom-to-top" with hyphens

We have retained the original. If "bottom to top" were being applied as an adjectival modifier of a noun this change would have been appropriate, but in this case it would be incorrect.

- line 50: zone is not capitalized when in the plural form (same as formations, units, etc.); correct this throughout the manuscript. corrected throughout the manuscript

- line 50: delete "however", not needed. Reworded

- line 51: "defy simple categorization" - I don't know what this means and checked the references to try to find out too. reworded

- line 56: add "...that contain abundant plagioclase" to help the non-expert.

We do not understand this suggestion. As explained in the next sentence, the Critical Zone is defined as that portion of the RLS containing chromitite layers - plagioclase-rich rocks are not present everywhere in the Critical Zone, but do occur in the Upper Critical Zone. The explanation for the non-expert is in the sentence immediately following the one in which he wants us to provide it.

- lines 57 and 58: use of the word "component" - I don't know what this means? Has it been defined? There must be clearer terms to use. **reworded to avoid the incorrect use of the word "component"**

- paragraph lines 48-60: most "Zones" should be "zones". **done**

- line 59: as argued here, the Upper Zone is the youngest part of the RLS, but that is contradicted later in the manuscript? Need some clarity here.

We are not aware of having suggested otherwise anywhere else in the manuscript. The Upper Zone does indeed appear to be the youngest part of the RLS, given that it cross-cuts all other units.

Page 3

- line 61: give abbreviations for the different chromitites as soon as you use them - Lower Group (LG), etc. **reworded**

- line 63: the UCZ did not fall (or walk, run, jog) - use "is defined by" **done**

- lines 67-69: I wrote "can be tested with geochemistry" on my marked up copy here - does this statement invalidate all existing geochemical studies of the chromitites? **Intrusion-wide geochemical comparisons of occurrences of individual named units have been done rarely or not at all except for the Merensky Reef and UG2 units, which we acknowledge to be correlatable across the entire complex. Scoon and Teigler (1994) made it clear that such correlations are questionable (cited reference 21) under the LG6 horizon as we state in the text. We are unaware of any more recent work that has been done to test the validity of the correlations.**

- line 72: this is the first time I have seen the use of "interburden" - I know what it means now because I looked it up and found that it is a mining term, but I suggest replacing it with a word that most readers will understand. **replaced with "intervening thickness"**

- lines 73-74: multiple use of "everywhere" - avoid repetition and **rephrase** one part of the sentence. **This is not repetitive. The point we are making is that in all places (i.e., everywhere) north of Burgersfort the UG3 is separate from the UG2, but that in all places (i.e., everywhere) south of that point the two occur together without any intervening layers. What other word could be used to replace "everywhere"? In this case we repeat the word intentionally for emphasis**

- line 77: yes, Bushveld Complex, but everywhere else it is written as RLS? **That is correct. the RLS is part of the Bushveld Complex; the Bushveld complex occurs as several different lobes.**

- lines 77-79: awkward sentence - try **rephrasing**. The subject gets lost. Avoid the word "clearly" as it assumes the reader intuitively knows what the authors have observed. **Rephrased and shortened to avoid getting bogged down in controversy about the relationship between the North and other lobes of the Bushveld Complex**

- lines 77-86: this is all one paragraph - right now, these are two small incomplete paragraphs. **We have**

put the first of these (now shortened) paragraphs into the preceding one, but left the second one alone.

- line 82: there is a problem here. The authors indicate that the Platreef and the Merensky Reef are one and the same based on apparent correlations across the Bushveld Complex. This is, however, far from accepted. The packages contain distinctively different rock types. I am unaware that clear continuity has been determined between the Northern Limb and the other lobes, both from geological and geophysical evidence. Field relationships defined for the Platreef are not relevant to the Merensky Reef. We disagree with this statement but to avoid unnecessary controversy we have removed or softened our stance. However, we do note that recent work by Kinnaird (her chapter of the book Platinum-group element (PGE) mineralization of the Bushveld Complex, South Africa, 2015, published by the Council for Geoscience) has traced the units of the Eastern and Western lobes into the Northern lobe with considerable confidence, and they have published maps that show the Merensky reef and the UG2 as distinct and mappable units well into the Northern lobe, into the Grasvalley area. Where the Merensky reef is mapped as overlapping onto the floor of the complex north of there it is named the Platreef. Since this work presents the most detailed map available of the Northern lobe, we consider it to be authoritative and to supersede any previous discussion of uncertainties which themselves were written in the absence of such detailed map-scale information. However we don't want to get off-track here so we have inserted the word "tentatively" and removed the declaration that the Main Zone must be older than the Merensky Reef.

- line 89: need to provide location information for these two boreholes (lat/long, depth drilled, location on map, etc.) We have indicated their locations on the map in Figure 1 and have given their lat/long coordinates in the text.

- line 91: friends are "encountered" in a bar, lithologies are not "encountered" by a drill hole. reworded as "intersected"

Page 4

- line 92: replace "bottom" with "base" done

- line 93: replace "entirety of" with "entire" done

- line 94: replace "encounter" - see line 91 above. reworded as "intersected"

- line 94: as written, the sentence indicates that the gabbronorite is 730 m in thickness - a small rewrite with clear this up. done

- line 95: where are the other chromitites? A small comment here might be useful in the context of these being sills. We now say that the hole was stopped at MG2. Therefore any chromitite lower than this would not have been intersected.

- line 97: "... is comparable with similar profiles from slightly farther east" - I'm not sure what this means, what profiles? Just a re-wording needed. done

- line 106: replace "interbedded" with "interlayered"
- line 107: replace "millimetric laminations to decametric beds" with "that range from mm to dm".
- lines 108-109: I cannot tell which photo this is? The reader needs some help here (even the specialist does, too). The notion of partial melting has not yet been brought up, so this catches the reader by surprise. Explicit references to individual panels of Figure 4 have been added in the main text to direct the reader's attention, and we have elaborated slightly on this first mention of partial melting by alluding to the old idea of a basal thermal aureole beneath the UG1 chromitite.
- lines 110-113: this is not a paragraph, just a single sentence that shows up out of the blue. put in previous paragraph and reworded.
- line 112: delete "clearly" done
- line 114: start with "Whole rock" "whole rock" is implicit in the fact that we name rock types and also in the word litho geochemistry.
- line 118: replace "litho geochemical data" with "geochemical results" "litho geochemical data" is more specific than "geochemical results" so we stand with the text as originally worded. For that matter, even the geochronological data are "geochemical", so the word has little value in this context.
- line 121: hyphen needed after "dilution" done
- line 122: add "weighted mean" before "²⁰⁷Pb/²⁰⁶Pb ages" done
- lines 114-119: there is something critical missing here for the reader and for future workers - we need a clear indication of what was sampled and analyzed for U-Pb geochronology? Where exactly was each sample taken from with respect to each chromitite? In the middle, above, below? Are the dating samples actually chromitites or the rocks in the package? This needs to be spelled out in the supplementary information with short statement here. I suggest indicating how far away individual samples from upper and lower contacts of chromitites were taken in centimeters. To make this easier for the reader to understand we have added some text referring to the various places where this information is supplied. We had already provided everything the reviewer asks for here, but it was difficult to sort it out from the various supplementary files. These data were supplied as Supplementary Information (text descriptions), Supplementary Figure 1 (photographs of dated core samples) and Supplementary Figure 2 (X-ray maps of polished thin sections of dated samples). We have added precise indications of the depths in hole for all of the core fragments included in each dated sample in response to the questions about where exactly each sample came from - comparison of this information with the summary log should be informative. We have also added to the summary log (Supplementary Data Table 1) an indication of where each dated sample was taken highlighted with red text.

- line 123: "multiple, reproducible, concordant U-P data" is awkward - try a rewrite.

We added the word 'and' before "concordant" to make it less awkward, but have kept the three descriptors as we consider each to be valid and necessary

- line 134: add "... and the results are plotted on..." at the end of the sentence. done

- line 135: "single crystals or up to 5 fragments of zircon grains" - need to be very clear about this in the data table (add a column), which specific analyses are single grains and which are multi-grain fractions? Are all baddeleyite analyses done on single grains as well?

We have added a column to our U-Pb data table that lists the number of grain fragments.

- line 136: "207Pb/206Pb weighted means" is incorrect - should be "weighted mean 207Pb/206Pb dates"
In this context the wording is correct as written (but will revise if requested by the Editor)

- line 137: is there any specific data collection or data reduction software used in addition to Isoplot?

We have inserted two sentences here to inform on the data acquisition software used, as well as the data reduction and age calculation software.

- line 137: should be "age results", not "data" - the data are in the analytical table. done

- line 139: ages don't "fall", people do (or branches from trees). reworded

- line 139: need to add "Ma" after each date.

- line 139-140: "in good agreement with the recent published age range" - I don't know what this means? What constitutes "good agreement"? It looks like they may be a little younger - certainly the Merensky Reef ages do not overlap. We have reworded this to indicate an overlap between our range of ages and Zeh's range of ages, but highlighted the fact that our Merensky Reef age is slightly but resolvably younger than their age for its immediate footwall.

- line 143: replace "data" with "results"; need "sample" after Main Zone

- lines 143-145: awkward phrasing - try a rewrite. The sentence reads that the age is discordant, but that is incorrect - the U-Pb results from this one fraction are discordant. I also can't tell which fraction this is on Fig. 5A? Try a different shading. We have adjusted the sentence and use a different shading on the fraction (data point) in question.

- line 147: "to ensure rigorous evaluation of the data" - I think this is overstated, you are just reporting two different age calculations. reworded without the offending phrase

- lines 133-148: there is no mention of the baddeleyite dates - are these not useful or important? And I guess there are some rutile dates somewhere too based on the title of the data table?

We have added a sentence after line 148, and had originally referred to the baddeleyite data after line 163, so now the baddeleyite results are referred to in two places.

- lines 133-148: the U-Pb results are excellent - very fine work that will be very useful for future workers. What about the reversely discordant results for the zircon (single grains or multi-grain fractions?) from the Merensky Reef sample? Everything else behaves so well. These are different - why?

The precision required to resolve the layering controversy in this study (~0.01%) is an order of magnitude greater than what the gold standard has been for the past 10 years or so in U-Pb geochronological studies. To achieve this, all samples, with the exception of the Merensky Reef sample, were analysed the same way using multiple Faraday collectors in static mode, which required a sizeable beam to obtain sufficient Pb and U. The Merensky Reef sample yielded fewer zircon grains to work with, especially after the HF etch treatment, and these were relatively low in U concentration. This meant that 3 of the 4 fractions of zircon for this sample had smaller emissions of Pb and U and were therefore analysed on our Daly detector system in dynamic, peak-hopping mode. These 3 data do indeed plot above the Concordia curve but, we must emphasize, they are still concordant as they plot well within the decay constant uncertainties (represented on the concordia plots by the grey band) and the final ages of the individual analyses are unaffected by this. The 4th point was analysed in static mode on multiple Faraday collectors and it plots just below the curve and gives an identical $^{207}\text{Pb}/^{206}\text{Pb}$ age as the data obtained using the Daly system. There are several possible reasons for the slight reverse discordance (U dead time of the measuring system, U peak shape, U blank, a possible minor interference on ^{205}Pb peak, for example). Ultimately, the slight reverse discordance has no effect on our final ages because we are using the $^{207}\text{Pb}/^{206}\text{Pb}$ ages not Pb/U ages.

- line 149: should be "U-Pb zircon age differences"; also, it's not just between the units, it also includes the Main Zone? **reworded**

Page 6

- line 155: I don't understand the definition of the "the most robust conclusion" here - the statistical analysis does not provide conclusions, just what is permissible. **We disagree. Our analysis leads us to conclude with 99% confidence that the MG2A Unit was emplaced after the overlying UG1 Unit. This is a minor semantic issue and if the editor wishes us to rephrase then we will.**

- line 157: replace "our age" with "the age" **done**

- line 159: replace "Myr" with "Ma" **done**

- lines 156-161: no spaces before %, should be "90%" **done**

- line 162: "but is nevertheless strongly suggestive" - suggestive of what? This is an unfinished statement. **added "that the Merensky Reef is younger than the Main Zone and therefore might be hosted by a sill "**

- line 163: replace "in agreement" by "identical within uncertainty" **done**

- line 164: replace "the zircon data" with "the U-Pb zircon results" **done**

- line 165: replace "Our data" with "Our U-Pb results"; replace "impossible" with "not possible"; add "that" after "accept" **done**

- line 166: delete "as being the" and replace with "resulted from the" **done**

- line 167: replace "suggest" with "propose" **done**

- lines 170-172: well-phrased and clear statement.

- lines 172-173: "the ultramafic intrusions formed due to assimilation, within deep conduits, of continental crust by komatiite magma" - as written, this is incorrect and misleading. The ultramafic intrusions formed from the crystallization of komatiitic magma, derived by high extents of partial melting of the mantle, which was contaminated by continental crust during transport from the mantle to the uppermost part of the crust. **reworded**

- line 177: delete "types of first order" - unnecessary words. **done**

- line 177: "our hypothesis" - which specific hypothesis? There are a number of proposals in the preceding paragraph, however, it is unclear which one is being referred to? **we have changed "hypothesis" to "hypotheses"; however we really are testing one group of ideas that cannot be separated from each other because they describe the entire process**

- line 178: what were the water contents and redox states of crystallization and partial melting? These need to be stated explicitly either here or in the supplementary table to allow researchers to reproduce the results. **We have added the requested information " with bulk H₂O content of 0.1 wt% and oxygen fugacity fixed at the value of the fayalite-magnetite-quartz solid oxygen buffer (i.e., FMQ;"**

- line 179: delete "generic" - not necessary **done**

- line 180: add "upper continental crust" - I got confused here as continental crust is intermediate in composition, not felsic **we have replaced the word "felsic" with "intermediate-felsic" (compared to ultramafic rocks, we thought that SiO₂ content of 65% seemed felsic, but indeed it is truly intermediate-felsic) Intermediate would be less than 63% SiO₂ and felsic would be greater than 68% SiO₂ by some definitions.**

- line 182 and 185: repetition of "comprises" - find another word for one of them or re-arrange a sentence. **Comprises is the right word to use for our intended meaning so it needs to be used twice. Sometimes words have to be used more than once on a page, to wit, the word "the".**

- line 183: magmas don't fall, ski, hop, or run - the temperature decreases reword According to the dictionary, the word "fall" certainly can be used to connote a decrease. We choose to retain the original wording.

- line 183: "felsic crustal rocks" - the average silica content of upper continental crust is ~66 wt%, so it's better to refer to "intermediate-felsic" here; note that this composition needs to be included somewhere in Supp Table 2 for reference. We now refer to intermediate-felsic. The composition used has been added to Supplementary Table 2.

- line 184: replace "which" with "that" as the word modifying "mass" and not "forms". done

Page 7

- line 185: not sure "felsic" is appropriate here for reasons stated above. reworded to intermediate-felsic

- lines 185-187: the decimal points being quoted can't be significant - just round off for the larger numbers. The notion of significance has little meaning here because these are not measurements - they are model values. We prefer to retain the same number of decimal points in the larger numbers as in the smaller numbers for consistency, unless the editor feels strongly that we should not.

- line 187: replace "matching" with "..., which matches". done

- line 188: avoid using "clinopyroxene-porphyratic pyroxenite" - not a good usage for an ultramafic cumulate; delete "well" - not necessary. "clinopyroxene-porphyratic pyroxenite" is the terminology widely used by workers in the RLS, to distinguish those pyroxenites having clinopyroxene crystals much larger than the much more abundant but smaller orthopyroxene crystals. We could call it clinopyroxene-megacrystic but that would just be adding another term to an already murky local terminology. Deleted "well"

- line 189: add B1 magma composition to Table 2 for reference and comparison. Done.

- line 190: replace "data" with "results" done

- lines 191-192: "performed a similar exercise" is awkward - rephrase. reworded to show that Eales did much the same thing using the same software and similar parameters

- line 193: incorrect usage of "which" - here it modified "conduit", but I don't think the conduit settled anywhere! This has been corrected by adding a comma to put ", resulting from crustal contamination of komatiite in the conduit," into a separate adjective phrase modifying "mush" thereby allowing "which" to apply correctly to "mush".

- lines 195-198: well-stated and clear.

- lines 198-201: expelling of melt to form Marginal Zone - interesting proposal, but the implications are that there are being expelled over 100s of km strike length, unless you develop a method for forcing them downwards through crystalline rock. You have introduced a mechanically challenging process in trying to solve another. Why not just mix the melt back into the slurry during sill emplacement? We are not sure what the reviewer means by this query. We are suggesting that the entire layer (UG2 Unit, for example) is a sill that extends all the way to the edge of the RLS, where it spread out into the surrounding sedimentary basin to form flanking sills now recognized as the Marginal Zone. During the process, partial melts of floor and roof rocks were added to the through-going magma. We have added some text to try to clarify our ideas.

- line 202: start new sentence with "This means that the chromite..." to avoid a very long and difficult to follow sentence. done

- line 203: where did the number "25 times greater" come from? It shows up without any context or support. We have removed the numerical estimate, preferring to leave this to a subsequent paper where it will be possible to present full supporting models and arguments.

- lines 2005-210: this is very complex and convoluted sentence that needs to be broken into pieces and reassembled as two separate sentences. Reworded.

- line 211: "flame structures" - why do they have to have a sedimentary origin? Why could they not result from the proposed partial melting process? They do. Flame structures result from the deposition of a dense sediment layer (e.g., chromitite) on top of a pre-existing deformable substrate (e.g., plagioclase crystal mush) that is less dense.

- line 215: delete "must" done

- line 216: replace "which" with "that" done

Page 8

- line 217: "repeated intrusion of sills before earlier sills" - what sills are we talking about here? This is in direct reference to one paper, but many other authors have proposed non-sill origins for the Bushveld Complex. reworded to separate into two distinct sentences addressing two different ideas.

- line 220: replace "using" with "and use"

- line 221: "thermoelasticity" of what? Thermoelasticity is that branch of physics that deals with stress and strain induced by changes of temperature in elastic solids.

- line 221: "Upper Critical Zone" - this is modeled as cold rock (573 K) - what happens when it is warmer? Since our ages differ by hundreds of thousands of years, we assume that the previously emplaced sills will have had time to cool. Using higher temperatures would result in smaller modeled stresses.

- line 222: what is a "magmatic pulse" - humans have a pulse, but do magmas? removed
- line 224: delete hyphen between "recently emplaced" - no hyphen when the word ends in "y". done
- lines 228-233: too long and complex - difficult for the reader to follow. Break into smaller bites. done
- lines 221-240: this entire section where the details of the modeling are explaining could be moved to a supplementary file, which would allow the authors to expand on the geological significance and issues of emplacing these sills (or to produce a shorter article). More space is used on the background to the modeling here than on the U-Pb geochronology, which is the foundation of the study. This section is not really very long, and we feel strongly that it is important. The reason we feel this way is that everyone we talk to about our ideas objects to the remarkably parallel disposition of the sills but then is mollified when they see the thermoelastic model - failing to address this issue in the main text would weaken our arguments significantly. If the editor wants this removed then we will add it to the Methods section.
- line 238: replace "Kyr" with "ka" done
- line 239: sentence is convoluted here with no punctuation - try putting brackets around "(200 m thick)" or re-arrange sentence. removed "200 m thick"

Page 9

- lines 248-252: clear statement - critical for the reader!

- lines 253 and 256: repetition of "strongly" - find another word for one of them. removed the first one
- lines 262-264: reference to younger ages reported in Zeh et al. (2015 EPSL) - yes, but their ET 2Ga synthetic solution result is 1 Ma younger than what your results give? It is difficult to know the reasons for the difference in age between our respective datasets for the 2 Ga synthetic solution. In communicating with other labs that have dated these solutions (there are also 0.5 Ga and 0.1 Ga solutions available through the Earthtime project) some have noted scatter in the data, which we speculate may mean possible heterogeneities, or condensates, in the solutions, and we are aware that some labs no longer analyse the solutions due to these problems. Whether heterogeneities exist and whether they may produce slight age bias in datasets has not been established, for now it is speculation; however, tests in our lab indicated that when the solutions were placed in Teflon dissolution capsules in an oven overnight at 200C, then aliquoted and analysed, the results were more reproducible in very small sample loads.
- We note that in the Zeh et al. study, the reported dataset was much larger than ours (N=23 vs our N=6), which could potentially result in age bias.
- line 262: there are no published ages for samples from the Lower Zone in Zeh et al. removed "Lower Zone"
- line 262-263: "numerous isolated compartments" - no, all the Zeh et al. samples from the CZ appear to

be from one compartment in the northeast Eastern Limb with the exception of one UG2 sample from the Western Limb. **reworded to avoid the unintended implication that Zeh et al. had dated multiple compartments.**

- lines 267-271: good statement, however, I think the modeling results need to address some of the realities of the layered sequences in the Bushveld Complex. The stress model involves lateral expansion parallel to the σ_1 - σ_2 plane. The layered rocks in the Bushveld Complex (RLS) are not necessarily flat, but curved with layered packages pitching out around dome structures that formed during emplacement and cooling of the hot cumulates (see excellent papers on structural geology of emplacement of the Bushveld Complex by Clarke et al 2005 SAJG, 2009 Lithos). The proposed mechanism is "simple", which is always appealing, but needs some modification in light of the geology. **All we have done here is to show that parallel emplacement is likely to be the rule rather than the exception. We have added a sentence allowing that real contact relations will be more complicated than this.**

- line 272: add "Bushveld Complex" here **done**

Page 10

- line 278: **replace** "data" with "U-Pb dating results"

- line 279: **delete** "tend to" - not necessary

- lines 280-281: "PGE deposition" - the PGE (as elements) are actually deposited? **They are deposited within sulfide liquid and chromite.**

- lines 284-287: when I read this statement, I expected such a wide-reaching statement to be supported by many studies - consider other supporting references. **We are not aware of other studies that made similar arguments. Here we are urging people not to place all unobserved magmatic processes in "magma chambers" or "staging chambers" which appear conveniently in a great many papers without any attempt to justify their existence. If the editor or the reviewer can suggest other references we will be pleased to include them.**

- lines 280-289: good statements and strong finish. I couldn't help but notice that a similar passage was written at the end of a recent review paper on the geochronology of layered intrusions in the "Layered Intrusions" book published by Springer (see Chapter 1, p. 59-60):

"By combining these advances in technologies and applications with our current capabilities of determining high-precision crystallization and cooling ages from a range of rock types in layered intrusions, Archean to Cenozoic, the next 10-15 years of research will undoubtedly provide many important, and unexpected, discoveries based on the geochronology of layered intrusions with implications for the evolution of mafic magmatism in the Earth's crust. The geochronological tools are now available to address whether major layered intrusions are assembled in discrete magmatic episodes, whether all layered intrusions are simple stratigraphic sequences of cumulates with the oldest

rocks at the base and the youngest at the top near the roof, whether discontinuities in cumulate sequences can be identified and time gaps measured, and whether their associated mineral deposits are directly related to the crystallization of their immediate host rocks or formed or were modified at some temporally resolvable time after crystallization. Answers to these questions will directly impact our understanding of the emplacement, crystallization, and cooling of these exceptional bodies of igneous rocks."

An acknowledgement that others are working on this fascinating and fruitful avenue of science appears to be warranted. This work had already been cited as reference 15, but another citation is added here, along with a direct quote "geochronological tools are now available to address whether major layered intrusions are assembled in discrete magmatic episodes, whether all layered intrusions are simple stratigraphic sequences of cumulates with the oldest rocks at the base and the youngest at the top"

References

Good use of references in general.

- is reference #8 to Scoon and Mitchell at the LASI Conference an abstract? If so, it cannot be used. removed, with regret - is it true that abstracts cannot be cited? These authors deserve credit for sticking their necks out, and for having been the first in recent years to come out and say this publicly - otherwise we seem to be taking that position ourselves, which is not strictly true (albeit, we were unaware of their position when we did our study, only learning of their work afterwards when we were ready to submit our paper).

- starting on page 13, lots of journal names are not italicized - double-check these. done

- need superscripts in Jaffey reference (line 403) done

- need total pages for book references in #49 and #50

- the Muller & Pollard reference is cited twice (#53 and #54) removed the first instance

Methods

- line 497: should be "major element oxide" done

- line 502: need to state what these reference materials are - either here or in the table. Hopefully some of them are mafic rocks? These are now listed. Yes several are mafic rocks.

- I assume that LOI was not determined by ICP-MS? added

- line 504 needs to begin with "For U-Pb geochronology, ..." as this is new analytical section done

- line 535: great to see a reporting of the EarthTime 2 Ga synthetic solution. I'm surprised that you make

no comment about the difference between your results and those from Zeh et al. (2015) - there is a 1 Ma difference between them, which is significant given the age differences you are working with. In the text, you indicate that the two studies give similar results, but you don't address this bias. **Please see above for lines 262-264**

- line 537: why a specific value of Th/U magma of 4.2? I think I know, but the non-specialist (anyone who does not know the details of Bushveld geochemistry) will not. **A first order correction for disequilibrium between mineral and magma is generally made by most (all?) workers who report U-Pb data. Most labs use an average Th/U crustal value of between 4 and 4.2, which is the generally accepted crustal average (originally published by Allegre). This is to correct for excess ^{206}Pb from the decay of ^{230}Th in the crystallizing zircon. Our Th/U value can be considered appropriate in light of the fact that in this study our model liquid Th/U composition is 4.4 (see Supplementary Table 2), and the Th/U of the cumulates is 4.12 (n=300). It is more important to make this correction in much younger samples, has only a very minor effect in older samples.** **If the editor wants us to, we will add some wording to this effect.**

- line 538: give the numerical values for the decay constants here in addition to the reference. This makes it very clear what was used - just add them in brackets. **Added.**

Tables

- there is no explanation in the text why the two different values of $^{238}\text{U}/^{235}\text{U}$ are used and reported in this table. If no mention is made, then why report them? To my knowledge, the international geochronology community has not yet agreed to make the shift to the Hiess et al. (2012) value of 137.818 - even these authors do not recommend accepting it without further study. Here's a quote from Blair Schoene's excellent overview on "U-Th-Pb Geochronology" in the Treatise on Geochemistry (second edition) that outlines the current thinking:

Schoene (2014 Treatise on Geochemistry), p. 359:

"Using the value of 137.818 {plus minus} 0.045 suggested by Hiess et al. (2012) would change the calculated λ_{235} by $\sim 0.03\%$, though, as discussed in that paper (Figure 13), a single study with full traceability to SI units needs to be carried out before new values of λ_{235} are adopted for use in geochronology. Furthermore, increasing the absolute resolution of U-Pb geochronology beyond the 0.1% level and/or verification of the accuracy of the λ_{238} value from Jaffey et al. (1971) requires further counting experiments."

The authors have presumably used the lower value of $^{238}\text{U}/^{235}\text{U}$ as this is what Zeh et al. used (2015 EPSL). For comparison with all existing U-Pb datasets, the 137.88 value is still the reference value. Some clarification of the table and the protocol used are needed to avoid confusion. I recommend reporting dates in the manuscript with the 137.88 value and report the dates with 138.818 values for reference in the table if you wish (maybe in supplementary material?), but make this clear to the reader.

some text has been added to the caption of Table 1 to explain this better to the reader.

- under Rock Unit, flip the text around so that the unit is first, then the rock type (makes more sense).

done

- the title of the table is incorrect - it is not "for rocks from the Critical Zone", but for rocks from both the Critical Zone and the overlying Main Zone. **changed**

- the use of the word "population" in the footnote is incorrect. As used, the authors are indicating that there are multiple populations of zircon in their samples that they analyzed - this is not the case (I hope!). They are "analyses" or "results". **changed to "analyses"**

Figures

Figure 1

- need to add "Bushveld Complex" in the caption title. **done**

- need to indicate sources for the figure - the current statement is insufficient as it assumes that the reader can figure out which references are appropriate on their own. **added**

- the caption needs to indicate what the ages are - weighted mean $^{207}\text{Pb}/^{206}\text{Pb}$ dates. **done**

- the drill holes for this study need to be clearly indicated on the map - I see TF3, but not the other ones. **The two other holes are from the Modikwa Mine, which is already indicated and is stated in the text.**

- the various boxes (around legend, around scale, around lat/long, around middle of nowhere in lower-centre) are distracting and none of them need to be there. **deleted**

- in the Eastern Lobe, the font size for the different sectors is not the same and should be. **corrected**

- in the legend, I cannot tell the difference between the "Merensky Reef" and the "Middle Group" - same colour and patterns basically. And there is no indication what "Middle Group" refers to in the caption. **Middle Group is now called Middle Group chromitites; the Merensky Reef line is red (always was, but line weight was too light to see, now heavier)**

- the graph to the right should have tickmarks along the bottom axis to allow the reader to easily read off the ages of the lowermost samples. **added; also modified upper age axis for clarity**

- the caption needs to define the abbreviations used in the graph (BR, MZ, UG, MG) - only the specialist knows these. **done**

Figure 2

- I like the idea of this figure, but I must admit that it is hard to see much in the scanned cores (these are great, but at this scale, I only see some colour variations and not the "key textures" as indicated). I'm not sure this will survive any further reduction on the printed page. Perhaps focus on one part and expand the scale or simplify the figure, minus the scans, and include the scans in the supplementary material?

For this issue we ask the editor for guidance. It is true that on a printed page much detail may be lost, but we have drafted the figure at sufficient resolution that a pdf version can be zoomed to show a large amount of important detail. Should we include links to high-resolution images of the core samples in supplementary or on-line edition? We do want this graphical information to be readily available to the reader because we consider it to be an extremely important part of the story. It is the field observation that motivated the entire study. We have added a part b to this figure showing some of the same core photos much larger. This could go in the main text or in supplementary materials at the editor's discretion.

- the decimal zeroes can all be deleted (e.g., 295 rather than 295.00) as they are not significant at this scale and take up space. **done**

- in the caption, delete "detailed" and "key textures" as these are not viewable by the reader.

- commas are missing throughout the caption **corrected**

Figure 3

- great photos, but there are some issues. First, add need a thick black line around each photograph to make them pop out of the page. Second, the photos look very washed out - the colour settings and contrast needs to be reset (this can be clearly stated in the caption). Third, I can follow the different contacts (just!), but I don't think a non-specialist can, so I recommend trying a version where the major contacts are carefully traced with thin white lines. **All three suggested changes have been made.**

- the white boxes around the text in both panels are distracting - try without the boxes and use white font all around (increase size a little to make it contrast enough).

- the figure caption is poor. The caption needs a correct title as in "Photographs of ...". The sentence beginning is "Randomly placed drill holes..." is very awkward as written and hard to decipher - it needs a complete rewrite. The text for (B) does not provide the clear description that the reader needs to evaluate the photo.

- the location (level) of the mine in which these photos were taken needs to be stated. **We do not have a record of the exact location. It was in an access drift where potholing had brought the chromitite down below its usual level.**

- there is no scale given for either of these photos - a scale must be given for all photographs otherwise they are of little use to the reader. **added - also added to other photos or described in captions.**

Figure 4

- I suggest adding boxes around them to make them pop - this is especially important when the colour in the photo is nearly the same as the page (see panel A). **done**

- the labels (A, B, C, etc.) are too large and the white boxes too distracting. Try using small white font. **changed to white but left font size alone because they disappear if they are any smaller and also white**
- the caption needs to start with "Photographs of ...". **done**
- Dwars River is not spelled correctly. In South Africa, it is written as "Dwarsrivier", but the usage should be in the English form here. **changed throughout manuscript and figures**
- the caption for A is a fragment of a sentence that does not make sense. **Not sure what this means - A is the subject, shows is the verb.**
- line 592: delete "also" **done**
- line 593: "interlayered", not "interbedded" **done**
- line 598: spell out "metres" and finish the sentence (beneath G?) **done**
- after looking at the photos and reading the caption, I realized that it will be very difficult for the non-specialist to actually see what the authors are referring to in the photo. I suggest labelling the photos (could use rock name abbreviations?) to make this clear, otherwise they are just grey rocks with some bands and swirls. **done**

Figure 5

- I recommend copying and pasting the tickmarks to the opposite sides of the graphs - this really helps the reader determine the values for points/symbols that are near all sides of the graphs. **done**
- capitalize "Reef" in panel B. **done**
- I suggest labeling the CL image as panel G (it looks me a few re-reads to get this). **done**
- the title of the figure is incomplete: "U-Pb concordia diagrams showing...". **expanded**
- the use of the word "data" is incorrect in this caption - the data are in the table, these are the results plotted on the diagrams. **corrected**

Figure 6

- copy the tickmarks to the opposite sides of the graph for reference. **done**
- the title is incorrect and should state "Diagram of Al₂O₃ vs. MgO showing ...". **done**

- "data" are not shown - they are in the table. It should read "Whole rock compositions are shown..."
corrected

- the box in the upper left is too large and distracting - reduce the space between the three labels and it will look better. **done**

- I have a lot of trouble reading this caption and understanding how the diagram works - I should be able to do this with just the figure and the caption. **the caption has been modified and expanded**

- what does "UG2 is weighted average of TF3 intersection" mean? I think I know, but it's not clear.
explained better

- it's not a "black curved trend", it's a "curved black line". **corrected**

- the last statement does not make sense to me. Two of the whole rock compositions plot on the restite trend, so they cannot be derived by mixing. **this would be consistent with near-total melt extraction**

Two of the whole rock compositions plot outside the possible mixing vectors and so are inconsistent with the proposed model? **yes, slightly inconsistent, and we have tempered the wording by saying "much of the observed range". This is an exceedingly simple model being used to demonstrate that melting of norite produces anorthositic residue. It is not an attempt to duplicate precisely the compositions of rocks that undoubtedly underwent a variety of complicated and inter-dependent processes before solidifying. A full discussion of this subject would require a separate paper, which is something we hope to write in the coming year and submit to a specialty journal.**

Figure 7

- copy the tickmarks to the opposite sides of the graph for reference - need tickmarks for the elements along the x-axis. **done**

- tighten up the spacing between the labels in the legend - it will look better and less crowded in the upper right. **done**

- title is incorrect, should be "Primitive mantle-normalized trace element concentrations for ...".
corrected

- line 618: delete "actual", substitute with "rock analyses" **done**

Figure 8

- the "old sill" text covers a line - find a way to avoid this. **corrected**

- the first sentence in the caption is awkward, especially "...sill 200 m thick 1,000 years and 100,000 years..." - too many numbers. Try a rephrasing to help the reader. **rephrased**

- consider changing the temperature from K to {degree sign}C - intuitively simpler. redrawn using °C

Supplementary Data Figure 1

- in the caption, depths in the drill hole need to be given for each core in the photo - don't make the reader try to find this information in the associated supplementary material done

- there are no scales again - absolutely need scales. the diameter of the core is now stated in the caption

Supplementary Data Figure 2

- need depths indicated in the caption for each sample shown done

Supplementary Material

Supplementary Information - sample descriptions:

- as a stand-alone document, this needs to convey some additional information to the reader (some of which is in the main text), including location of the drill hole, total depth, angle, and width of core. This has been added

- need a hyphen in "a medium-grained..." done

- change "faint igneous lamination" to "weak lamination" - presumably defined by the orientation of plagioclase crystals? Or not? Yes. added

- "orthocumulate texture" done

- replace all usages of "intercumulus" to "interstitial" to avoid any future issues of interpreting terminology. done

- when using "the rock is fine grained ...", there is no hyphen - see other usages on page ? we have hyphenated all instances of fine-grained and medium-grained.

Supplementary Table 1:

- needs a careful edit and some rewriting. As presently written, the Descriptions look like what was written on first observing the core - some tightening up of language would be useful as this table will be archived and available to future workers. some modifications made, but since the core is no longer available for inspection it seems unwise to replace observations made at the time with new wording that might be inaccurate.

- watch out for capitalization issues in headings **fixed**
- Lower CTC column heading is not defined (what does this mean for the reader?) **spelled out as "lower contact"**
- Unit Name abbreviations are not defined. **Spelled out Merensky Reef, Bastard Reef. Chromitite unit names are thoroughly explained in the text, and nobody but specialists will look at the summary log, so everyone will know what these units are.**
- some measurements in metres have two decimal places, others have none - need to be consistent. **fixed**
- please avoid the \$ sign for sulphides - use "S" **done**
- under Colour Codes, what does "mafic interburden" mean? **replaced "interburden" with "rocks"**

Supplementary Table 2:

- title: what kind of "compositions"? **rock compositions i.e., lithochemical compositions i.e., whole rock analyses.**
- need to indicate what reference materials were used? **given in the Methods section in the main text. If the editor wants we can supply some QA/QC data including a number of replicate analyses of reference materials. The present study does not rely on extreme fidelity of geochemical data so we are not sure how much care needs to be taken to document the quality of these routine whole rock analyses.**
- lines 62-67: where do these values come from? Need to indicate how they were calculated? **asterisks and a footnote have been added to indicate that these are model values calculated using alphaMELTS as described in the main text.**
- decimals issue in "Depth in hole" column **fixed**
- need to indicate in footnote what methods were used to determine the major element oxide and trace element concentrations? **Done, and give reference to the Methods section**
- decimal usage issue throughout table - what is shown is what the authors are indicating is significant, but it is variable? **We report data with the number of significant figures provided by the laboratory.**
- column Q, Cr₂O₃ in wt%? There are values for leuconorites to anorthosites that are 81%, 405%, 96% - these are not possible? 405%? Move the Cr₂O₃ column to the left to just after Al₂O₃ where it belongs. **fixed**
- column R, Cr (ppm): need to remove decimal places in lines 22-24 and 40-46 (not significant). **fixed**

- in the REE, watch out for significant decimal places (some apparently have none). **fixed**
- decimal places in Yb-U under model compositions are all wrong - need to correct. **fixed**
- line 59, column 2: I don't understand the label (Thiel Well; Ta, Sr, U interpolated) -make a clearer statement in a footnote **done**

Supplementary Table 3:

- the title says that there are rutile analyses, but none are reported? I assume these will be part of a follow-up paper? **Correct. The reference to rutile has been removed**
- there are no U and Pb concentrations reported? These need to be included. **These have been added.**
- there is no mass of the fraction reported? **These have been added.**
- footnotes should clearly indicate the decay constants used (even if they are in the main text), as well as data reduction programs, and a reference to Mattinson (2005) for the chemical abrasion pretreatment protocol. The supplementary tables need to work as stand-alone items for future reference. **Decay constants now appear in the text and in the footnotes of Suppl Table3; Mattinson has been cited in reference to chemical abrasion of zircon grains.**
- why is the assumed Th/U in the magma a very specific value of 4.2? I think I know, but it would be useful for the reader to know. **See response above.**
- the footnotes need some kind of symbology to reference specific parts of the table - very useful for the non-specialist. **Not sure what "symbology" is required---each parameter/ratio/abbreviation has been described in the footnotes.**
- a separate column needs to be added to indicate how many grains were used for each fraction. **This has been added.**

Supplementary Table 4:

- title should include "Bushveld Complex", so that future users will know what this table refers to. **added**
- there are column headings missing (columns B and C) **added**
- footnotes need symbols to guide the reader. **done**
- it's not clear what the different sub-sections of this table are? For the reader to navigate the calculations, each needs a clear heading **done**

- what does "using Z1-Z5..." mean? I know, but definitely not the non-specialist who is trying to figure out how these important calculations were made. **rephrased, and a reference to the discussion in the main text has been added.**

Reviewer #2 (Remarks to the Author):

This is, by and large, a well-written paper describing new geochronologic data, aimed at establishing age relations between different layers of the Bushveld Intrusion (or intrusions). The goal is to test a hypothesis that these different layers might be formed by a succession of discrete intrusions rather than sequential formation from a large chemically evolving magma body. Generating age data of sufficient precision to be able to distinguish the crystallisation ages of different layers in such old rocks is presumably quite a big technical challenge. It certainly looks (figure 1) as though some "age discrepancies" are present (figure 1) although perhaps one could debate whether this is enough evidence alone to support the hypothesis. However, these data are also supported by a number of field / textural observations which plausibly mostly support the interpretation being proposed. I have not seen the rocks myself in the field, although I've read about them. It is obviously a difficult job to summarize in a short paper the particular observations in an overall context of a huge amount of published work on the Bushveld. I did not always find it easy to keep track of the various descriptions of the rock units which are not immediately familiar to me (lines 48-86, for example). Finally, the paper develops the hypothesis in two quite worthwhile ways 1) modelling the melting and mixing relations from the point of view of chemical composition and mineralogy, and 2) some thermomechanical modelling to underpin the observational work with some physical ideas.

To summarize the paper is therefore quite ambitious. By and large it is very dense - there is quite a lot to assimilate given the short length. The thermomechanical modelling alone, for example, one could easily imagine developed as a paper in its own right, rather than as here, where it goes by a little fast, to be frank. These ideas seem generally plausible to me, but to be really convinced, I'd probably rather see something more detailed. Overall I found the paper rather dense, but nevertheless, the different kinds of reasoning and measurements also give a quite a nice balance to the approach.

I am basically supportive of publication with no significant changes. I don't know whether it will be found convincing by all readers who know a lot more about the Bushveld than I do, and who may be inclined to evoke counter arguments / observations, but I found it sufficiently thought-provoking and well argued enough to be a valuable contribution.

REVIEWER COMMENTS:

Reviewers' comments:

Reviewer #1 (Remarks to the Author):

I wrote a review of a previous version of this manuscript for Nature Geoscience and had about 18 pages of comments. The authors have made significant progress in organization, presentation, and writing that now allow the very interesting and important science of the study to really stand out. Below, my general comments are mostly repeated from the earlier review, but updated where needed based on changes in the present manuscript. Additional suggestions are also provided for polishing the text, tables, figures, and supplementary material for final presentation.

This is a fascinating contribution on one of the landmark intrusions of petrology, the Bushveld Complex, a mafic-ultramafic layered intrusion that has influenced many of our basic concepts of how igneous bodies crystallize and how magmatic ore deposits of chromium, platinum group elements, and vanadium form. The authors report new U-Pb zircon dates for rocks mostly from the Critical Zone of this intrusion, the largest mafic-ultramafic layered body on Earth. They use relatively small differences in age between the samples to propose that the constituent layers do not form in a regular fashion from bottom-to-top, but are in fact out-of-sequence, which implies that some/many layers were emplaced as sills into pre-existing cumulates. Evidence of emplacement as sills is based on inferred field relationships (provided as photographic evidence) and the textural-chemical fingerprint of partial melting of host gabbro-norite that produced very thin residues of contact anorthosite. From this, the authors speculate on the physical emplacement mechanisms of these sills, which may extend laterally for 100s of km, and provide a first-order thermomechanical model to explain how new sills should follow pre-existing sills and how each successive sill should be emplaced beneath consolidated sills. The end result is a novel view of the emplacement and crystallization of layered intrusions that is markedly different from the conventional view of progressively crystallizing magma chambers.

The strength of the study lies in the deft combination of high-precision U-Pb dating, field relationships, and modeling results. The U-Pb geochronological results are excellent and will be extremely useful to future workers as they continue to unravel the complexities presented in dating of the Bushveld Complex. There is a great deal more dating that will need to be done on rocks of the Bushveld Complex to fully appreciate the timescales of emplacement, crystallization, and cooling - with only the results for 5 samples presented in this study, focused almost entirely on chromitite-bearing horizons, there are significant gaps in sampling from an intrusion that locally exceeds 8000 metres in thickness. The alphaMELTS modeling of partial melting of host gabbro-norite to produce an anorthosite residue nicely ties observations to a predictable model that can be tested with more detailed studies in the future. The thermomechanical modeling of sill emplacement provides first-order resolution to the apparent observations of decreasing age with increasing depth in the Upper Critical Zone and will go a long way to convincing skeptics who would prefer the more traditional view.

Specific Comments

Abstract: very good - clear and concise with impact

Line 25: replace "intrusive complexes" with the simpler "intrusions"

Lines 33-37: awkward phrasing. Try a rewrite or split into two sentences. The "which" refers to RLS but really should refer to ages. If you replace with "that" then there are three "that" usages within about 10 words. This is a key sentence, so you want to get it right.

Lines 49-86: a long and very important paragraph - excellent summary of field constraints on stratigraphic sequences. Do you want to address the new paper by Latypov (2016, Journal of Petrology) on field constraints for the Merensky Reef? Latypov argues strongly (very strongly) that the Merensky Reef must represent the floor to a large, progressively crystallizing magma chamber. Your results clearly indicate a different story.

Line 98: should be "World Heritage Site"

Line 105: brackets around (e.g., MG4A and MG4B)

Line 118: space in "of textures"

Lines 119-123: is this supposed to be two sentences? There is something funny going around the part of "in the Burgersfort area".

Line 158: space in "of the"

Line 173: "show that"

Line 183: delete " ,"

Lines 186-199: good argumentation

Line 200: not sure the hyphen is needed in "sill-emplacment hypothesis"?

Line 228: space in "expelled laterally"

Lines 238-240: awkward sentence - try a small rewrite

Line 257: delete extra period

Lines 298-301: well-stated summary

Lines 302-313: good, strong paragraph

Lines 314-335: excellent finish

Line 596: consider a replacement for "achieved U-Pb geochronological data" - seems awkward.

Line 612: add "in the Western Lobe" after TF3 to make it immediately clear where the drillhole is located

Line 614: these crosses are hard to see on the figure?

Line 614: Need abbreviation definitions for LCZ, UCZ, MZ

Line 618: for clarity, write as "both upper and lower contacts of the UG2 Unit"

Line 619: the figure shows UG3/UG2, but this is not specifically noted in the caption.

Line 621: I had trouble guessing which were the "critical contacts" on the figure?

Line 632: should be "World Heritage Site" with capitals

Line 636: suggests using quotes for "migmatitic"

Line 652: Diagram showing Al₂O₃ vs MgO for whole rocks from....? Could use something specific here.

Line 666: Same as previous comment on title. "...natural and model rocks" related to?

References

- comprehensive - lots of important and long-lost papers are cited

Methods

Line 371: great to know the mass discrimination correction - is there an uncertainty available for this value?

Line 372: is there a value for SRM-982 that can be reported?

Tables

- good - very useful to report the dates as calculated with ²³⁸U/²³⁵U of 137.818 and 137.88. This is going to be an issue for the next few years until the community settles on a single value. The difference is larger than the measured age differences reported in the study, which is why this is so important.

- consider adding spaces on either side of {plus minus} in the table so that the dates do not look like one long number - it's hard to compare them without seeing the uncertainty part added as well

- should be "UG1 pyroxenite" with a space

Figures

Figure 1

- the Lobe labels look very large on this figure compared to the other font. Maybe try reducing the pt. size just a little and also centre justifying them to make them float better within the overall box.

- remove box around scale as it adds too many distracting lines in this part of the map

- move "Dwars River" and "Modikwa Mine" text out into the white areas and indicate where they are with arrows - very hard to see them on the figure

- spell out "Oliphants River"

- add "meters" for the Depth part of figure on the right

- copy x-axis numbers from top and place on bottom of graph to help the reader with the ages for the samples down here

- avoid having text over top of lines (see BV561 and dashed line)

Figure 2

- I still wish the core photos were larger, especially compared to the colored logs on the left of each panel.

- good idea to include the blow-ups - probably should move this to supplementary material. There is a scale in the caption, but a direct scale on the figure would be more effective. What about adding some labels to help the reader with interpretation? There is space between the cores where rock types and contacts could be identified.

Figure 3

- indicate "1 m" on each photo directly rather than having the reader search for the scale in the caption

Figure 4

- the figure caption should indicate what the gray line represents - uncertainty in concordia related to uncertainty in the decay constants for U

- the big bubbles for the ages seem a little overwhelming compared to the actual data as reported by the small ellipses. Why not remove the bubbles and just place the ages closer to the small dots in the centre of concordia?

- just a warning that my print-out on a high-res photocopier lost the light gray of the concordia curve. Maybe darken this up a little.

Figure 7

- tickmarks on the x-axis would be useful

- filling the white symbols for the model cumulate and model B1 and having the dashed lines sent to back would make a better looking figure

Supplementary Data Figure 1

- good

Supplementary Data Figure 2

- good

Supplementary Material

Supplementary Information - sample descriptions:

- good

- line 29: replace "collecting" with "and collected" to indicate past tense

- if the total weights of the composite samples that were crushed are available, this would be good information to include for future workers (i.e., how much material is needed to successfully extract zircon from these kinds of rocks)

Supplementary Table 1:

- good

Supplementary Table 2:

- make all font sizes equal (for example, 11 pt)

- just need a heading for "Th/U" on rows 58 and 64

Supplementary Table 3:

- good

Supplementary Table 4:

- good

- give common Pb composition used for 2.05 Ga in footnote

Reviewer #2 (Remarks to the Author):

In my original review, I did not ask for any specific revisions. However, I did hint that the paper is trying to do an almost impossible job: in a short paper, trying to summarize their geochronologic results in the context of a huge literature on the Bushveld, as well as describing a new thermomechanical model. The authors gave no reaction whatsoever to my comments, but obviously responded to a huge number of detailed comments from the other referee. My impression is that the efficiency of the writing has improved with respect to the previous version. Some of the descriptive material about the Bushveld, which was heavy to wade through, seems clearer to me this time round. It seemed to me that asking the authors to explain more details of the model etc. would be impossible, so I decided to take a broad view. Personally I find that papers which require vast quantities of supplementary material difficult. Overall, nevertheless, I think this works quite well as a possibly controversial, thought-provoking contribution.

The justification for publishing the paper in a broad audience journal is its positioning in a long-standing discussion about how "layering" may be produced in different intrusions. So it may be useful if I just pick up here on part of what I get as the "take home message". By and large, despite many attempts at physical modelling, a good "universal" model of a layering mechanism within a closed body of cooling magma - whether it be crystal settling, or thermochemical convection, etc. has not really emerged it seems to me. Which leaves "magma chamber replenishment" type mechanisms open - in other words the layering must be attributed to some kind of episodicity in the supply of magma to the intrusion, which does indeed raise the question of what the state of a magma body is when the

next "batch" arrives.

The current paper, is basically appealing to the same kind of episodicity as a potential explanation of layering, but postulating sill behaviour in terms of physical emplacement mechanism as the main novelty. Invoking sills to explain layering is not new, but the thermo-mechanical model is, I think. This is ok, but there is a chemical side to things, and I do think the authors have made a reasonable attempt, to look at a "contaminated-komatiite" petrologic model, which is interesting.

Apart from the economic geology aspect of the chemistry and petrology, I think the fact that abundant orthopyroxene is predicted by the MELTS simulations is promising, because the Bushveld and other similar age intrusions do have this mineral in abundance, which is not necessarily the case in younger ones. So perhaps the association with komatiites is plausible for such old intrusions, but does this restrict the generality of the model?

Hence, my final reaction is that perhaps the authors might add a short statement on the potential generality. My guess is that the basic sill mechanism can have broad potential to explain some of the layering in a range of intrusions, but presumably the magma composition, thermal structure, scale (ie. overall volume of magma), episodicity etc. will influence results. There remains a, perhaps secondary, "chicken and egg" type problem, in that this mechanism does require that a sill already be present in order to influence the next one by the thermo-elastic effects it creates. So why does the first sill form? The model does not solve the problem of why the magma intrudes rather than extrudes in the first place, but it does say why, once the process starts, that a layered structure might naturally develop. This is useful.

Reviewers' comments:

Reviewer #1 (Remarks to the Author):

I wrote a review of a previous version of this manuscript for Nature Geoscience and had about 18 pages of comments. The authors have made significant progress in organization, presentation, and writing that now allow the very interesting and important science of the study to really stand out. Below, my general comments are mostly repeated from the earlier review, but updated where needed based on changes in the present manuscript. Additional suggestions are also provided for polishing the text, tables, figures, and supplementary material for final presentation.

This is a fascinating contribution on one of the landmark intrusions of petrology, the Bushveld Complex, a mafic-ultramafic layered intrusion that has influenced many of our basic concepts of how igneous bodies crystallize and how magmatic ore deposits of chromium, platinum group elements, and vanadium form. The authors report new U-Pb zircon dates for rocks mostly from the Critical Zone of this intrusion, the largest mafic-ultramafic layered body on Earth. They use relatively small differences in age between the samples to propose that the constituent layers do not form in a regular fashion from bottom-to-top, but are in fact out-of-sequence, which implies that some/many layers were emplaced as sills into pre-existing cumulates. Evidence of emplacement as sills is based on inferred field relationships (provided as photographic evidence) and the textural-chemical fingerprint of partial melting of host gabbro that produced very thin residues of contact anorthosite. From this, the authors speculate on the physical emplacement mechanisms of these sills, which may extend laterally for 100s of km, and provide a first-order thermomechanical model to explain how new sills should follow pre-existing sills and how each successive sill should be emplaced beneath consolidated sills. The end result is a novel view of the emplacement and crystallization of layered intrusions that is markedly different from the conventional view of progressively crystallizing magma chambers.

The strength of the study lies in the deft combination of high-precision U-Pb dating, field relationships, and modeling results. The U-Pb geochronological results are excellent and will be extremely useful to future workers as they continue to unravel the complexities presented in dating of the Bushveld Complex. There is a great deal more dating that will need to be done on rocks of the Bushveld Complex to fully appreciate the timescales of emplacement, crystallization, and cooling - with only the results for 5 samples presented in this study, focused almost entirely on chromitite-bearing horizons, there are significant gaps in sampling from an intrusion that locally exceeds 8000 metres in thickness. The alphaMELTS modeling of partial melting of host gabbro to produce an anorthosite residue nicely ties observations to a predictable model that can be tested with more detailed studies in the future. The thermomechanical modeling of sill emplacement provides first-order resolution to the apparent observations of decreasing age with increasing depth in the Upper Critical Zone and will go a long way to convincing skeptics who would prefer the more traditional view.

Specific Comments

Abstract: very good - clear and concise with impact

Line 25: replace "intrusive complexes" with the simpler "intrusions"

done

Lines 33-37: awkward phrasing. Try a rewrite or split into two sentences. The "which" refers to RLS but really should refer to ages. If you replace with "that" then there are three "that" usages within about 10 words. This is a key sentence, so you want to get it right.

We have split this into two sentences and reworded slightly

Lines 49-86: a long and very important paragraph - excellent summary of field constraints on stratigraphic sequences. Do you want to address the new paper by Latypov (2016, Journal of Petrology) on field constraints for the Merensky Reef? Latypov argues strongly (very strongly) that the Merensky Reef must represent the floor to a large, progressively crystallizing magma chamber. Your results clearly indicate a different story.

We are reluctant to engage with just one recent example of the literally hundreds of papers that have been written based on the presumption that the layers all formed at the base of a large open magma chamber. We note that Latypov's work was primarily an exposition of the notion that hot new magma had partially eroded the floor of the magma chamber followed by what he calls "*in situ*" crystallization on the contact, a process that is not in any way excluded by our model of sill injection by hot magma. Therefore his work does not contradict ours in any significant way. However if the Editor feels strongly about this we can add a citation and some text to explain why we don't think his work is important in this context.

Line 98: should be "World Heritage Site"

corrected

Line 105: brackets around (e.g., MG4A and MG4B)

corrected

Line 118: space in "of textures"

corrected

Lines 119-123: is this supposed to be two sentences? There is something funny going around the part of "in the Burgersfort area".

corrected

Line 158: space in "of the"

corrected

Line 173: "show that"

corrected

Line 183: delete ", "

corrected

Lines 186-199: good argumentation

Line 200: not sure the hyphen is needed in "sill-emplacement hypothesis"?

removed

Line 228: space in "expelled laterally"

corrected

Lines 238-240: awkward sentence - try a small rewrite

done

Line 257: delete extra period

done

Lines 298-301: well-stated summary

Lines 302-313: good, strong paragraph

Lines 314-335: excellent finish

Line 596: consider a replacement for "achieved U-Pb geochronological data" - seems awkward.

changed to "acquired"

Line 612: add "in the Western Lobe" after TF3 to make it immediately clear where the drillhole is located

done

Line 614: these crosses are hard to see on the figure?

enlarged

Line 614: Need abbreviation definitions for LCZ, UCZ, MZ

added

Line 618: for clarity, write as "both upper and lower contacts of the UG2 Unit"

done

Line 619: the figure shows UG3/UG2, but this is not specifically noted in the caption.

added

Line 621: I had trouble guessing which were the "critical contacts" on the figure?

added some labels and lines marking contacts and rock types

Line 632: should be "World Heritage Site" with capitals

done

Line 636: suggests using quotes for "migmatitic"

added

Line 652: Diagram showing Al₂O₃ vs MgO for whole rocks from....? Could use something specific here.

lengthened the title

Line 666: Some as previous comment on title. "...natural and model rocks" related to?

lengthened the title

References

- comprehensive - lots of important and long-lost papers are cited

Methods

Line 371: great to know the mass discrimination correction - is there an uncertainty available for this value?

We do not assign an error to the Daly mass discrimination correction. It is a systematic error (internal error) that affects all the data systematically.

Line 372: is there a value for SRM-982 that can be reported?

The NBS (National Bureau of Standards) SRM-982 values and ratios are easily accessible (comes out at the top of a Google search) and generally not cited in each paper. Those who use the standards will know the values, and those who do not can look them up readily. If the editor desires it we can list the values.

Tables

- good - very useful to report the dates as calculated with $^{238}\text{U}/^{235}\text{U}$ of 137.818 and 137.88. This is going to be an issue for the next few years until the community settles on a single value. The difference is larger than the measured age differences reported in the study, which is why this is so important.

- consider adding spaces on either side of {plus minus} in the table so that the dates do not look like one long number - it's hard to compare them without seeing the uncertainty part added as well

done

- should be "UG1 pyroxenite" with a space

done

Figures

Figure 1

- the Lobe labels look very large on this figure compared to the other font. Maybe try reducing the pt. size just a little and also centre justifying them to make them float better within the overall box.

done

- remove box around scale as it adds too many distracting lines in this part of the map

done

- move "Dwars River" and "Modikwa Mine" text out into the white areas and indicate where they are with arrows - very hard to see them on the figure

done; also added a line in the Legend for "Locality described in text"

- spell out "Oliphants River"

done

- add "meters" for the Depth part of figure on the right

done

- copy x-axis numbers from top and place on bottom of graph to help the reader with the ages for the samples down here

done

- avoid having text over top of lines (see BV561 and dashed line)

shortened lines

Figure 2

- I still wish the core photos were larger, especially compared to the colored logs on the left of each panel.

The core photos are larger in the second part of this figure, which we agree should be in the supplementary material and should be archived at the full resolution of the images we supply to the journal to permit the reader to enlarge them on screen. The textures are critically important.

- good idea to include the blow-ups - probably should move this to supplementary material. There is a scale in the caption, but a direct scale on the figure would be more effective. What about adding some labels to help the reader with interpretation? There is space between the cores where rock types and contacts could be identified.

labels and scale added

Figure 3

- indicate "1 m" on each photo directly rather than having the reader search for the scale in the caption

Figure 4

- the figure caption should indicate what the gray line represents - uncertainty in concordia related to uncertainty in the decay constants for U

- the big bubbles for the ages seem a little overwhelming compared to the actual data as reported by the small ellipses. Why not remove the bubbles and just place the ages closer to the small dots in the centre of concordia?

The large ellipses show the actual uncertainty in the position of the indicated age on the concordia. We feel that this is useful information to retain.

- just a warning that my print-out on a high-res photocopier lost the light gray of the concordia curve. Maybe darken this up a little.

done

Figure 7

- tickmarks on the x-axis would be useful

done

- filling the white symbols for the model cumulate and model B1 and having the dashed lines sent to back would make a better looking figure

done

Supplementary Data Figure 1

- good

Supplementary Data Figure 2

- good

Supplementary Material

Supplementary Information - sample descriptions:

- good

- line 29: replace "collecting" with "and collected" to indicate past tense

- if the total weights of the composite samples that were crushed are available, this would be good information to include for future workers (i.e., how much material is needed to successfully extract zircon from these kinds of rocks)

Supplementary Table 1:

- good

Supplementary Table 2:

- make all font sizes equal (for example, 11 pt)

done

- just need a heading for "Th/U" on rows 58 and 64

done

Supplementary Table 3:

- good

Supplementary Table 4:

- good

- give common Pb composition used for 2.05 Ga in footnote

Reviewer #2 (Remarks to the Author):

In my original review, I did not ask for any specific revisions. However, I did hint that the paper is trying to do an almost impossible job: in a short paper, trying to summarize their geochronologic results in the context of a huge literature on the Bushveld, as well as describing a new thermomechanical model. The authors gave no reaction whatsoever to my comments, but obviously responded to a huge number of detailed comments from the other referee. My impression is that the efficiency of the writing has improved with respect to the previous version. Some of the descriptive material about the Bushveld, which was heavy to wade through, seems clearer to me this time round. It seemed to me that asking the authors to explain more details of the model etc. would be impossible, so I decided to take a broad view. Personally I find that papers which require vast quantities of supplementary material difficult. Overall, nevertheless, I think this works quite well as a possibly controversial, thought-provoking contribution.

The justification for publishing the paper in a broad audience journal is its positioning in a long-standing discussion about how "layering" may be produced in different intrusions. So it may be useful if I just pick up here on part of what I get as the "take home message". By and large, despite many attempts at physical modelling, a good "universal" model of a layering mechanism within a closed body of cooling magma - whether it be crystal settling, or thermochemical convection, etc. has not really emerged it seems to me. Which leaves "magma chamber replenishment" type mechanisms open - in other words the layering must be attributed to some kind of episodicity in the supply of magma to the intrusion, which does indeed raise the question of what the state of a magma body is when the next "batch" arrives.

The current paper, is basically appealing to the same kind of episodicity as a potential explanation of layering, but postulating sill behaviour in terms of physical emplacement mechanism as the main novelty. Invoking sills to explain layering is not new, but the thermo-mechanical model is, I think. This is ok, but there is a chemical side to things, and I do think the authors have made a reasonable attempt, to look at a "contaminated-komatiite" petrologic model, which is interesting.

Apart from the economic geology aspect of the chemistry and petrology, I think the fact that abundant orthopyroxene is predicted by the MELTS simulations is promising, because the Bushveld and other similar age intrusions do have this mineral in abundance, which is not necessarily the case in younger ones. So perhaps the association with komatiites is plausible for such old intrusions, but does this restrict the generality of the model?

Hence, my final reaction is that perhaps the authors might add a short statement on the potential generality. My guess is that the basic sill mechanism can have broad potential to explain some of the layering in a range of intrusions, but presumably the magma composition, thermal structure, scale (ie. overall volume of magma), episodicity etc. will influence results. There remains a, perhaps secondary, "chicken and egg" type problem, in that this mechanism does require that a sill already be present in

order to influence the next one by the thermo-elastic effects it creates. So why does the first sill form ? The model does not solve the problem of why the magma intrudes rather than extrudes in the first place, but it does say why, once the process starts, that a layered structure might naturally develop. This is useful.

we agree that the reason why sills form is not addressed in our work, but note that others have published explanations that we find satisfactory, one of which we have cited (Gudmundsson). To address the broader implications of our work we have added the following paragraphs in the conclusions:

Our proposal that macrolayering in a layered intrusion might result from the emplacement of a series of individual sills at successively lower levels need not be restricted to the Bushveld Complex but could also be applied other well-known layered intrusions. The existence of intrusive upper contacts on the sills is very easy to overlook due to the absence of recognizable chilled margins on these hot erosive boundaries as illustrated in Supplementary Figure 1.

The notion of the "magma chamber" is so deeply ingrained in current thinking about igneous petrology that it is invoked in countless papers without any attempt at justification. Here we have shown that much of the layering in one of the world's truly iconic layered intrusions need not be explained in terms of processes occurring in an open melt-filled chamber. Invisible and undetected "deep-seated magma chambers" and "staging chambers" appear throughout the current literature, but the notion of a "staging chamber" beneath the Bushveld Complex or any other complex magmatic system may also be unnecessary since the composition of many batches of incoming crystal-laden magma can be modeled simply by taking a common Proterozoic magma type (komatiite) and adding some melted wall-rock to it as it rises through the continental crust along a flow-through conduit. We therefore urge some cautious reflection before appeal is made to the prevalent concept of vast open chambers filled with essentially crystal-free melt except in those cases where direct evidence can be seen for their existence. Although small magma chambers like the famous Skaergaard Intrusion of east Greenland¹⁸ have doubtless existed at some times and places, their primacy as the model for formation of large layered intrusions must now be critically re-examined in each case.